# Deep Incentive Design with Differentiable Equilibrium Blocks

**Vinzenz Thoma** [1] [†]   **Georgios Piliouras** [2]   **Luke Marris** [2]

## Abstract

Automated design of multi-agent interactions with desirable equilibrium outcomes is inherently difficult due to the computational hardness, non-uniqueness, and instability of the resulting equilibria. In this work, we propose the use of game-agnostic *differentiable equilibrium blocks* (DEBs) as modules in a novel, differentiable framework to address a wide variety of incentive design problems from economics and computer science. We call this framework *deep incentive design* (DID). To validate our approach, we examine three diverse, challenging incentive design tasks: contract design, machine scheduling, and inverse equilibrium problems. For each task, we train a single neural network using a unified pipeline and DEB. This architecture solves the *full distribution* of problem instances, parameterized by a context, handling *all* games across a wide range of scales (from two to sixteen actions per player).

## 1. Introduction

Game-theoretic equilibria provide us with the language to describe and study multi-agent interactions. Many works have focused on their computation and the respective complexity classes. In contrast, from a policy-making perspective, we typically want to solve the *inverse* problem of setting up the rules of a game to guarantee desirable equilibrium outcomes. In economics, for example, institutions design taxes or markets with the aim of improving social welfare or revenue at the anticipated equilibrium behavior. Similarly, as we create agentic AIs interacting with each other, our aim is to set up incentives such that the resulting behavior is aligned with societal welfare.

This fundamental problem of *incentive design* (ID) can be

[†]Research conducted while employed as an intern at Google DeepMind. [1]Department of Computer Science, ETH Zurich & ETH AI Center, Switzerland [2]Google DeepMind. Correspondence to: Vinzenz Thoma <vinzenz.thoma@ai.ethz.ch>.

*Proceedings of the 43rd International Conference on Machine Learning*, Seoul, South Korea. PMLR 306, 2026. Copyright 2026 by the author(s).

formalized as a mathematical program with equilibrium constraints (MPEC) (Luo et al., 1996). In the upper-level problem, an incentive designer selects decision parameters $\theta$ to minimize a loss function $\mathcal{L}$. These parameters $\theta$ induce a general-sum normal-form game $G$. In the lower-level problem the players face the induced $G$ and respond by playing an equilibrium (Eql) $\sigma^*$. This equilibrium in turn impacts the designer's loss $\mathcal{L}$. Crucially, rather than solving a single, isolated instance, we aim to *learn a design policy that generalizes* across a whole class of problems, parameterized by a *context* $\omega \sim \Omega$. The problem is thus learning $\theta$, such that the induced games $G(\theta; \omega)$ yield equilibria $\sigma^*$ that minimize the designer's loss $\mathcal{L}_{\sigma^*}(\theta; \omega)$ in expectation over $\omega$.

---

**The Incentive Design Problem**

$$\min_{\theta} \mathbb{E}_{\omega \sim \Omega} \left[ \mathcal{L}_{\sigma^*}(\theta; \omega) \right] \quad \text{s.t. } \sigma^* \in \text{Eql}(G(\theta; \omega)) \quad \text{(ID)}$$

---

As an example, the context $\omega$ may define a base game $B$, and the learnable parameter $\theta$ defines a conditioned and possibly constrained perturbation $\delta$ to induce the game $G(\theta; \omega) = B(\omega) + \delta(\theta; \omega)$. The loss function in this case may be negative welfare $\mathcal{L}_{\sigma^*} = -\sum_{p,a} \sigma^*(a) G_p(a; \theta, \omega)$, where $G_p(a; \theta, \omega)$ is the payoff of each player.

We focus on correlated and coarse correlated equilibria as the lower-level constraints. It is a natural choice for a problem setting where a central designer exists and can help correlate players' strategies. Moreover, mathematically, it allows us to select the unique maximum-entropy equilibrium from the convex equilibrium set, rendering this choice *differentiable* with respect to $\theta$.

Recently, Marris et al. (2022) and Liu et al. (2024) presented two distinct approaches to train neural networks to map a wide spectrum of normal-form games to their unique maximum entropy (coarse) correlated equilibrium. Once trained, they allow the approximate computation of the equilibrium's derivatives in any game up to a certain size. We thus refer to such networks as *differentiable equilibrium blocks* (DEBs). In this work, we introduce the framework of *deep incentive design* (DID), where we parameterize $G(\cdot; \cdot)$ as a class of (game-theoretically equivariant) neural networks, called *mechanism generators*, and learn the network's weights $\theta$ to minimize the expected loss over the possible distribution of

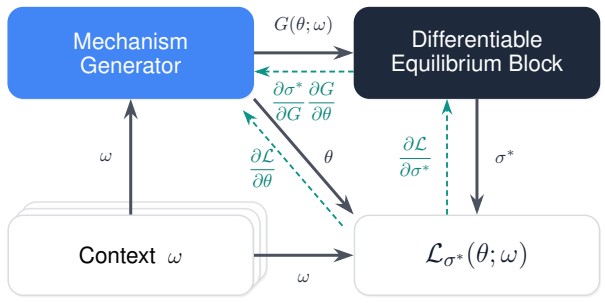

*Figure 1.* Deep Incentive Design framework. Black (green) arrows show the forward (backward) computation.

the context $\omega$, which is passed as input to the network. To achieve this, we compose the mechanism generator with a pretrained DEB to evaluate $\mathcal{L}$ at the predicted equilibrium $\sigma^*$ and then backpropagate through the DEB to train the generator weights $\theta$. Our training pipeline is illustrated in Figure 1.

**Our Contributions** Our contributions are threefold:

- On a conceptual level, we introduce the DID framework, a principled and general approach to solve (ID) by backpropagating through a DEB.

- From a system perspective, we present a highly scalable, modular and generalizable training pipeline. Our networks are trained as *mechanism generators*: they take as input the context $\omega$ and can handle the whole class of problems in $\Omega$ instead of being retrained for individual contexts. The networks have a *game-theoretically equivariant* architecture, which (i) provides a strong inductive bias by respecting domain symmetries, (ii) enables principled dimensionality reduction, and (iii) allows the networks to train on games of different shapes. Specifically, we train a single network for all games of sizes ranging from $2\times2$ up to $16\times16$.

- Experimentally, we show the promise of DID by tackling several well-studied and diverse problems from the literature, including multi-agent contract design (Holmstrom, 1982), machine scheduling (Heydenreich et al., 2007) and inverse equilibrium problems (Kuleshov & Schrijvers, 2015).

## 2. Related Works

We situate our approach in the related literature and refer the reader to Appendix B for an in-depth comparison.

### 2.1. Gradient-Based MPEC

Several works have studied gradient-based approaches to solve MPECs. Generally these can be grouped into two approaches: those using implicit differentiation, relying on the implicit function theorem, and those using automatic differentiation, unrolling the computational graph of some converging dynamics. Both approaches rely on assuming a unique Nash equilibrium exists. Implicit differentiation has been used to converge to unique Stackelberg equilibria (Fiez et al., 2020; Wang et al., 2022). Li et al. (2020) study (ID) for a single, fixed context and frame the lower-level equilibrium problem as a monotone variational inequality, which they solve with both implicit and automatic differentiation. Extensions to this work have included distributed algorithms (Grontas et al., 2024), hierarchy-free two-timescale methods (Liu et al., 2022; Li et al., 2023; Maheshwari et al., 2022), and using automatic differentiation to design congestion games (Li et al., 2024b), mean-field games (Corecco et al., 2025), and social-dilemmas (Baumann et al., 2018; Balaguer et al., 2022).

### 2.2. Differentiable Economics

Mechanism design has been described as "inverse game theory". It is concerned with designing the rules of a (Bayesian) game, specifically the allocations and payment rules, such that the players truthfully reveal their private preferences and the designer maximizes social welfare or revenue.

Conitzer & Sandholm (2002; 2004) were the first to propose automated mechanism design, i.e. the (computational) design of a tailored mechanism to the specific problem. More recently, Dütting et al. (2024) introduced *differentiable economics*, using neural networks to parameterize an auction mechanism. The network takes as input the bids and outputs the payments and allocations. Some constraints, such as payment non-negativity and allocation feasibility are hard-coded in the architecture, others such as strategyproofness are enforced via a Lagrangian penalty. Several works have since proposed improved training pipelines and architectures (Duan et al., 2022; Tacchetti et al., 2022; Ivanov et al., 2022; Rahme et al., 2021; Wang et al., 2024; Duan et al., 2023b; Curry et al., 2023; 2024). We refer to Curry et al. (2025) for a comprehensive survey of the field.

### 2.3. Reinforcement Learning and Incentive Design

The problem of incentive design has been studied in the context of reinforcement learning with respect to designing rewards in MDPs (Thoma et al., 2024; Chen et al., 2022; Chakraborty et al., 2024; Shen et al., 2024; Chen et al., 2024) and Markov games (Gerstgrasser & Parkes, 2023; Brero et al., 2024; Huang et al., 2024; Canyakmaz et al., 2024).

Zheng et al. (2022) proposed the idea of an *AI economist*, where multi-agent RL is used to simulate an economy and its outcomes under different tax policies, which was later extended by Curry et al. (2022). These approaches take a descriptive approach, using heuristics mimicking human behavior without equilibrium guarantees.

## 3. Preliminaries

### 3.1. Game Theory and Equilibria

In this work, we study the design of $N$-player *normal-form games*. Each player $p$ has an action space $\mathcal{A}_p$. Players choose their action simultaneously, resulting in the joint action $a = (a_1, \ldots, a_N) \in \prod_{p=1}^{N} \mathcal{A}_p = \mathcal{A}$. We denote by $a_{-p}$ the actions of all players, except $p$. A joint action $a$ yields a payoff $G_p(a)$ for player $p$. We use $G$ to refer to the game defined by the payoffs $(G_1, \ldots, G_p)$. A distribution $\sigma(a)$ over actions is called a *joint strategy* or simply *joint*. In case players choose actions independently, the joint factorizes as $\sigma(a) = \prod_{p=1}^{N} \sigma_p(a_p)$ into the players' *marginals*. For a given joint $\sigma$, each player receives the expected payoff $\sum_{a \in \mathcal{A}} \sigma(a) G_p(a)$.

The solution concept of $G$ is a joint $\sigma^*$, from which broadly speaking no player has an incentive to deviate. If any unilateral deviation gives a player at most an $\varepsilon$ gain in expected payoff, $\sigma^*$ is called an *$\varepsilon$-equilibrium*. If $\varepsilon = 0$, we simply call $\sigma^*$ an *equilibrium*. In this work, we discuss three equilibrium concepts, which are gradual generalizations of each other.

$\sigma^*$ is an *$\varepsilon$-correlated equilibrium* ($\varepsilon$-CE) (Aumann, 1974), if no player achieves more than an $\varepsilon$ gain in expected payoff by deviating to any action $a'_p$, after having received a recommended action $a''_p$, drawn from $\sigma^*$. More formally, $\sigma^* \in \varepsilon$-CE($G$), if for all $p \in [1, N], a'_p \in \mathcal{A}_p, a''_p \neq a'_p \in \mathcal{A}_p$:

$$\sum_{a \in \mathcal{A}_{-p}} \sigma^*(a''_p, a_{-p})[G_p(a'_p, a_{-p}) - G_p(a''_p, a_{-p})] \leq \varepsilon.$$

$$(\varepsilon\text{-CE})$$

In case players can only choose to deviate before observing the recommended action, we call $\sigma^*$ an *$\varepsilon$-coarse correlated equilibrium* ($\varepsilon$-CCE) (Moulin & Vial, 1978). Formally, $\sigma^* \in \varepsilon$-CCE($G$), if for all $p \in [1, N], a'_p \in \mathcal{A}_p$:

$$\sum_{a \in \mathcal{A}} \sigma(a)[G_p(a'_p, a_{-p}) - G_p(a)] \leq \varepsilon. \qquad (\varepsilon\text{-CCE})$$

Both equilibrium notions require a common source of randomness to act as a coordination mechanism. This need not be an explicit signal, but could be a shared history of play. In fact, in a repeated game setup, no-regret learning algorithms, such as regret matching, use this history to provably converge to the set of (coarse) correlated equilibria, further motivating their use in this work (Hart & Mas-Colell, 2000).

A (C)CE is also a *Nash equilibrium* (NE), if it further holds that the joint can be written as the product of independent player marginals, i.e. if the players choose their actions independently. Note that NE $\subseteq$ CE $\subseteq$ CCE and $\varepsilon_1$-(C)CE $\subseteq \varepsilon_2$-(C)CE for $\varepsilon_1 \leq \varepsilon_2$. As every normal-form game has a NE (Nash, 1951), it follows that none of the equilibrium sets is empty. Further note that the set of $\varepsilon$-(C)CEs always is a convex polytope, which is generally not true for NEs, with the exception of two-player constant-sum games when $\varepsilon = 0$. Our DID framework requires the equilibrium set being a convex polytope. However, the approach works analogously for $\varepsilon$-CEs and $\varepsilon$-CCEs.[1] We therefore write $\varepsilon$-Eql($G$) to denote the convex equilibrium polytope of $G$ for statements that hold equivalently for both $\varepsilon$-CEs and $\varepsilon$-CCEs.

While the NE is often used in the literature, it is not suitable for our purposes. In particular, there are several obstacles to tackling (ID) with Nash constraints via gradient-based optimization. These include the computational hardness (Daskalakis et al., 2009), the multiplicity of NEs and that for general-sum games the set of NEs is not necessarily connected. To add to that, as we vary $\theta$ certain connected components of the NE set may disappear and new ones appear (Kohlberg & Mertens, 1986).

### 3.2. Incentive Design

Incentive design refers to the problem of designing games or game interventions with the goal of minimizing some upper-level loss that depends on the resulting game's equilibria. It is a fundamental problem capturing many strategic interactions, where a leader acts—or commits to act—before a set of followers, and was first studied by Stackelberg (1934) in the context of duopolies.

In our setup, there is a designer that controls a continuous parameter $\theta \in \mathbb{R}^d$ for some $d \in \mathbb{N}$. The designer's objective is to minimize the expected loss $\mathbb{E}_{\omega \sim \Omega}[\mathcal{L}_{\sigma^*}(\theta; \omega)]$ over the context $\omega$, drawn from a distribution $\Omega$. The loss depends on $\theta, \omega$, and $\sigma^*$. The latter is the equilibrium solution of a general-sum normal-form game $G(\theta; \omega)$, whose payoffs $G_p(a; \theta, \omega)$ are parameterized by both the decision variable $\theta$ and the drawn context $\omega$. The designer thus needs to learn a $\theta$ such that, in expectation over $\omega$, the players playing the induced games $G(\theta; \omega)$ are incentivized to play the designer's desired joint. The resulting MPEC is written down in Equation (ID).

For notational brevity we drop the dependence of $\sigma^*$ on $\omega$ and $\theta$ throughout this paper. We assume that the function $\mathcal{L}_\sigma(\theta; \omega)$ is differentiable with respect to both the decision

---

[1]DID could also be extended to implementing other equilibria, as long as their equilibrium set is a convex polytope, such as communication equilibria or Bayes-correlated equilibria.

parameter $\theta$ and a given joint strategy $\sigma$ for all $\omega$.[2] We further assume that the payoffs $G_p(a; \theta, \omega)$ are differentiable for all $a$ and $\omega$ with respect to $\theta$.

# 4. Deep Incentive Design

In this section we formalize our framework. We show that we can select a unique and differentiable equilibrium from the $\varepsilon$-Eql set. The differentiability is pivotal, allowing us to cast (ID) as a machine learning problem.

## 4.1. Differentiating through the Equilibrium

In this work, we use $\varepsilon$-Eql($G$) as the lower-level solution concept. It is a natural choice as the incentive designer can serve as a coordination mechanism for the players. Moreover, this choice yields two advantages. First, the polytope's constraints are linear functions of the payoffs and thus of $\theta$. Second, the set is convex, allowing us to define a strongly convex *equilibrium selection function* to choose a unique $\sigma^* \in \varepsilon$-Eql($G$), such that the choice is locally Lipschitz continuous and thus differentiable almost everywhere with respect to $\theta$ for $\epsilon > 0$.[3] In this work, we choose the unique maximum-entropy equilibrium $\varepsilon$-ME-Eql($G(\theta; \omega)$). This is motivated by the established theory on efficiently computing maximum-entropy equilibria (Ortiz et al., 2007; Marris et al., 2022). Following this approach we can reformulate (ID) in a more tractable form as

$$\min_{\theta} \ \mathbb{E}_{\omega}\left[\mathcal{L}_{\sigma^*}(\theta; \omega)\right] \ \ \text{s.t.} \ \sigma^* = \varepsilon\text{-ME-Eql}(G(\theta; \omega)),$$

$$(\varepsilon\text{-ME-ID})$$

where $\varepsilon$-ME-Eql($G(\theta; \omega)$) is the unique solution to $\max_{\sigma} H(\sigma)$ s.t $\sigma \in \varepsilon$-Eql($G(\theta; \omega)$) and $H(\sigma) = -\sum_a \sigma(a) \log(\sigma(a))$.

We established that the derivative $\frac{d\sigma^*}{dG(\theta;\omega)}$ exists if we choose $\sigma^* = \varepsilon$-ME-Eql. By the chain rule, this implies the existence of $\frac{d\mathcal{L}_{\sigma^*}(\theta;\omega)}{d\theta}$, which is equal to $\frac{\partial L_{\sigma^*}(\theta;\omega)}{\partial \theta} + \frac{\partial \mathcal{L}_{\sigma^*}(\theta;\omega)}{\partial \sigma^*} \frac{d\sigma^*}{dG(\theta;\omega)} \frac{dG(\theta;\omega)}{d\theta}$.[4]

Using the $\varepsilon$-ME-Eql ensures we can differentiate through the lower level of ($\varepsilon$-ME-ID) and enables us to use gradient-based optimization. However, computing the gradients exactly would require differentiating through a convex program with the constraints from ($\varepsilon$-CCE) respectively ($\varepsilon$-CE), which is impractical for large problems. Moreover, there

should be some generalization between the different contexts without having to rerun the optimization algorithm. This motivates a machine-learning based approach.

## 4.2. Incentive Design with Neural Networks

Having established the differentiability of $\mathcal{L}$, we can cast ($\varepsilon$-ME-ID) as a machine learning problem. We parameterize the function $\omega \mapsto G(\theta; \omega)$ as a neural network with weights $\theta$. We will refer to this network as the *mechanism generator*. This emphasizes that we do not learn a single mechanism to tackle a single task, represented by a single context. Instead, we train the network to learn the optimal game interventions for all $\omega \in \Omega$.

In order to train a mechanism generator, the network architecture requires a DEB that computes $\sigma^* = \varepsilon$-ME-Eql($G(\theta; \omega)$) on the forward pass and $\frac{d\sigma^*}{d\theta}$ on the backward pass. As mentioned in Section 1, both Liu et al. (2024) and Marris et al. (2022) have presented different approaches to train such DEBs for computing the $\varepsilon$-ME-Eql of any given game $G$ using an equivariant neural network architecture. They were able to solve very large normal-form games up to sizes of $64 \times 64$ for two players and $16 \times 16 \times 16$ for three players. While their work showed impressive results in the forward computation, the potential of backpropagating through their networks to approximate $\frac{d\sigma^*}{dG(\theta;\omega)}$ has been unexplored. Indeed, it is a priori unclear whether the gradients of these DEBs are sufficiently well-behaved to be useful for solving ($\varepsilon$-ME-ID) at scale.[5]

Figure 1 illustrates our DID pipeline, which composes the mechanism generator and the differentiable equilibrium oracle. The mechanism generator takes as input a batch of contexts $\omega^1, \ldots, \omega^B$ and outputs the induced games $G(\theta; \omega^1), \ldots, G(\theta; \omega^B)$, which are passed through the DEB to evaluate the minibatch loss. On the backward pass, we backpropagate through the DEB to train the mechanism generator's weights $\theta$. The DEB's weights are fixed, as it is pre-trained on predicting the $\varepsilon$-ME-Eql.

To summarize, with our DID framework and the use of DEBs, we are able to reduce the difficult game-theoretic problem (ID) to a standard optimization problem, amenable to the rich toolkit of machine learning. Notably, having a neural network trained across the whole context space $\Omega$ has two advantages over gradient-based optimization for a fixed context: (i) once trained, we can tackle many problems in parallel and quickly as the forward pass takes only $\mathcal{O}(|\mathcal{A}|)$ instead of having to rerun the optimization algorithm, and

---

[2] Note we do not directly assume that the function is differentiable with respect to the *equilibrium* joint $\sigma^*$, as this also varies with $\theta$. Instead, we show this explicitly in Section 4.

[3] The inequality is strict. For $\varepsilon = 0$ the set Eql may be a single point on the face of the simplex and not differentiable in $\theta$. For $\varepsilon > 0$, the differentiability almost everywhere follows from Rademacher's theorem.

[4] $dG(\theta; \omega)$ is short for the change in payoffs $(G_1, \ldots, G_N)$.

[5] This higher-order approximation problem of neural networks is well-documented in the literature and has led to novel training approaches such as *Sobolev Training*, using ground-truth gradients to improve the networks first-order approximations and sample efficiency (Czarnecki et al., 2017). However, in our setting ground-truth gradients are not available or too expensive to compute.

(ii) for any fixed $\omega$, ($\varepsilon$-ME-ID) is generally a nonconvex problem and gradient-based approaches easily get stuck in bad local minima. Using neural networks with high-dimensional parameter spaces and their ability to generalize between settings helps with escaping some of these bad local minima.

## 5. Experimental Results

In this section, we describe the setup and performance of DID on three hard problems from the literature: (i) multi-agent contract design, (ii) inverse equilibrium problems, and (iii) machine scheduling.

### 5.1. Network Architecture

For two of the three experiments–the inverse equilibrium and machine scheduling problems–the mechanism generators take as input the initial payoffs and output perturbed payoffs of the same shape $[N, |\mathcal{A}_1|, \ldots, |\mathcal{A}_N|]$.[6] These are of size $N \prod_{p=1}^{N} |\mathcal{A}_p|$, presenting a challenge for training on large games. Fortunately, the game-theoretic maps we are trying to learn have many *equivariances*. A function is equivariant to a permutation $\tau$ if it commutes with it, i.e. $f(\tau(x)) = \tau(f(x))$. In our case, any permutation of either the players or their action spaces in the input payoffs should also commute with the map $G(\theta; \omega)$, i.e. $G(\theta; \omega)$ is equivariant with respect to these permutations. Building a network architecture that respects these equivariances gives the network a strong inductive bias, as every input represents all possible $N! \prod_{p=1}^{N} (|\mathcal{A}_p|!)$ permutations of itself. As outlined below, an equivariant architecture also massively reduces the number of trainable parameters in the network, resulting in a more scalable training pipeline.

Our mechanism generators are built using variations of the following fundamental equivariant layer. In our experiments the data consists of a channel dimension $C$ and a set of equivariant dimensions (such as the player dimension of size $N$ or the players' strategy dimensions of size $|\mathcal{A}_p|$). To preserve these equivariances, we apply $I$ pooling functions $\phi_i$, whose output is concatenated along the channel dimension before being passed through a linear layer and an activation function $f$. The weights $w$ of the linear layer are shared between all points in the equivariant dimensions. At layer $l$, they just depend on the $C_{l+1}$ output channels and the $IC_l$ input channels. Therefore, the number of parameters depends neither on $N$ nor on $|\mathcal{A}_p|$, but only on the selected number of channels $C_1, \ldots, C_L$ and pooling functions $I$, where $L$ is the number of layers. Moreover, the networks can handle games of different sizes, allowing for even greater generalizability. The output $g_{l+1}(c_{l+1}, \ldots)$ of

the $l+1$-th equivariant layer can be recursively expressed as

$$f\left(\sum_{c_l=1}^{C_l} \sum_{i=1}^{I} w(c_{l+1}, c_l, i)\phi_i(g_l(c_l, \ldots)) + b(c_{l+1})\right).$$

For the contract design problem, the rationale is akin to above. The main difference is an additional equivariance with respect to permutations of the outcomes, resulting in an additional set of pooling functions applied along the outcome dimension. The exact architectures and training pipelines are described in Appendix A.

### 5.2. Experiment Setup

In our experiments, we solved ($\varepsilon$-ME-ID) with both $\varepsilon$-ME-CE and $\varepsilon$-ME-CCE as the lower-level solution concept, where $\varepsilon = 0.01$. Across all experiments we used the same two DEBs, one for computing the $\varepsilon$-ME-CCE and one for the $\varepsilon$-ME-CE, following the approach of Marris et al. (2022). The DEBs were trained to handle games with $N = 2$ players with action spaces between $2 \times 2$ and $16 \times 16$.

For all experiments, we evaluated the loss both at the approximate $\varepsilon$-ME-Eql computed from the DEB and the exact equilibrium (up to a tolerance) computed with the ECOS solver (Domahidi et al., 2013) using `cvxpy` (Diamond & Boyd, 2016; Agrawal et al., 2019) with default tolerances. This allows us to measure the errors from the DEB. Furthermore, we also compute a "polished" (local optimum) solution found using `scipy`'s (Virtanen et al., 2020) Nelder–Mead method (Nelder & Mead, 1965), initialized from the solution found by our approach. This method alone cannot identify a globally optimal solution in the high-dimensional parameter space.[7] However, initializing it from the found solution allows us to quantify the local improvement missed by DID and serves as an approximate local upper bound.

### 5.3. Multi-Agent Contract Design

Multi-Agent Contract Design was introduced by Holmstrom (1982). Consecutive works focused on the complexity of implementing equilibria in multi-agent contracts under different assumptions such as linear contracts or binary vs. general action spaces Babaioff & Winter (2014); Dütting et al. (2023; 2025b); Cacciamani et al. (2024); Castiglioni et al. (2023). Recently, Dütting et al. (2025a) introduced a model of multi-agent contract design where the lower-level problem is a (coarse) correlated equilibrium, similar to the $\varepsilon$-ME-Eql solution concept we use in this work. Despite the great economic interest in contract design, we are not aware of existing, implementable algorithms for the multi-agent case, underlining the promise of DID.

In multi-agent contract design we consider a set of agents

---

[6]In these cases, the context is equal to the initial payoffs. The details can be found in Sections 5.4 and 5.5 and Appendix A.

[7]Although it can be a good approximation of the global optimum in the game settings with low dimensionality, e.g. $2 \times 2$.

$N$, who are playing a normal form game with costs $c$, where each agent incurs a cost $c_p(a)$ for the joint action $a$. $a$ is mapped to an outcome $o \in O$, via a transition probability $P(o|a)$. The outcome is observed by the *principal* who receives a payoff of $W(o)$. Together the tuple of costs, transitions and principal payoffs defines the context of the multi-agent contract design problem. We write $\omega = (c, P, W)$ and use the superscripts $c^\omega, P^\omega, W^\omega$, when referring to the costs, transitions or payoffs of a specific context $\omega$. To increase the expected payoff, the principal offers a positive payment[8], called a *contract* $v_p(o; \theta, \omega) \in \mathbb{R}_{\geq 0}$ to each agent, contingent on the observed outcome. Importantly, the principal cannot directly observe the agents' actions nor condition contracts on these, which is referred to as *moral hazard*. The agents aim to maximize their expected payoff $G_p(a; \theta, \omega) = \sum_o P^\omega(o|a) v_p(o; \theta, \omega) - c_p^\omega(a)$ in the induced game $G(\theta; \omega)$, whereas the principal aims to maximize the payoff minus the total contracts paid $\sum_o P^\omega(o|a) \left( W^\omega(o) - \sum_p v_p(o; \theta, \omega) \right)$. Following Dütting et al. (2025a), who consider (C)CEs, we are using the unique $\varepsilon$-ME-Eql as the lower-level solution concept.[9] Our multi-agent contract design problem can thus be formulated as an instance of ($\varepsilon$-ME-ID) as follows:

$$\max_\theta \mathbb{E}_\omega \left[ \mathbb{E}_{a \sim \sigma^*, o \sim P^\omega} \left[ \left( W^\omega(o) - \sum_p v_p(o; \theta, \omega) \right) \right] \right]$$
$$\text{s.t. } \sigma^* = \varepsilon\text{-ME-Eql}(G(\theta; \omega)).$$

Consider the contract design problem illustrated in Figure 2 as an intuitive example. There are two players "Alice" and "Bob", who are siblings and tasked to set up a Christmas tree. They each have two actions, they can either help or not help set up the tree and will incur costs depending on their exerted efforts. Their choices stochastically lead to one of three outcomes: (i) the tree is set up if they either both helped or with a $50\%$ chance if one helped, (ii) the tree is set up wrong and burns down with a $50\%$ chance if only one child helped, and (iii) no tree is set up if no one helps. The father prefers the tree being set up over not being set up over the tree burning down. Importantly the father comes home and will not know who helped and who did not, i.e. he cannot observe the joint action $a$. He can only observe the tree, i.e. the outcome $o$ and promise to pay his children contracts contingent on the observed outcome.

To train our mechanism generator for square two-player

---

[8]The fact that the contract designer can only give but not take money away from agents is referred to as *limited liability*.

[9]The $\varepsilon$-ME-Eql corresponds to the point of the $\varepsilon$-Eql polytope closest to the center of the probability simplex. The principal's utility is a linear function and thus maximized at the vertices. The $\varepsilon$-ME-Eql thus never presents the maximum estimate of the principal's utility in the induced game, making the solution more robust to players playing other joints in the $\varepsilon$-Eql set.

games, we sample the size of the action space uniformly from $\{2 \times 2, \ldots, 16 \times 16\}$, the number of measurable outcomes uniformly from $\{2, \ldots, 10\}$, the transitions from joint actions to outcomes from a Dirichlet distribution with $\alpha = 0.1$, the principal payoffs $W(o; \cdot)$ uniformly from the interval $[-5, 5]$ and the agents' cost from the interval $[-4, 0]$. Note that, in contrast to the other two experimental settings, there are three inputs $c^\omega, P^\omega, W^\omega$ with respective shapes $[N, |\mathcal{A}_1|, \ldots, |\mathcal{A}_N|], [|\mathcal{A}_1|, \ldots, |\mathcal{A}_N|, O], [N, O]$. These are broadcasted and concatenated along the channel dimension to yield an input of shape $[3, N, |\mathcal{A}_1|, \ldots, |\mathcal{A}_N|, O]$. Nonetheless, the outputted contracts will be of shape $[N, O]$, requiring the use of different equivariant layers that at some point collapse the player and action dimensions. The details of this architecture can be found in Appendix A.

To evaluate the mechanism generator, we calculate the change in expected principal utility defined as

$$\mathbb{E}_{\omega, o \sim P^\omega} \left[ \mathbb{E}_{a \sim \sigma^*} \left[ \left( W^\omega(o) - \sum_p v_p(o; \theta, \omega) \right) \right] \right.$$
$$\left. - \mathbb{E}_{a \sim \sigma^0} \left[ \left( W^\omega(o) - \sum_p v_p(o; \theta, \omega) \right) \right] \right],$$

where $\sigma^*$ is the $\varepsilon$-ME-Eql under the game induced by the learned contracts $v_p(\cdot; \theta, \cdot)$ and $\sigma^0$ is the $\varepsilon$-ME-Eql under the initial game with costs $c_p^\omega$ and no contracts offered. For each game shape in $\{2 \times 2, \ldots, 16 \times 16\}$ and number of outcomes in $\{2, \ldots, 10\}$ we evaluated 8 randomly generated games. We then locally polished the solutions using 400 function evaluations. Our results can be found in Table 1a. For both CEs and CCEs, the learned $\theta$ produces contracts that consistently yield measurable improvements of the principal's utility over no intervention. While the found contracts perform well when evaluated under the DEB, we see a decrease in performance when evaluating with the exact solver (ECOS). This shows that the mechanism generator may be utilizing inaccuracies in the DEB. This could be ameliorated in future works by continuously training the DEB with the games generated by the generator. In all settings, except the CE being evaluated with ECOS, we see that the DID solution can be improved locally by no more than a factor of roughly 2.

## 5.4. Inverse Equilibrium Problems

Traditionally, there has been much attention on computing the equilibrium of a given game. However, there is the challenging inverse problem of coming up with a game that implements the desired equilibrium. This can be equivalently framed as rationalizing observed equilibrium behavior by defining suitable payoff functions and has been studied by Kuleshov & Schrijvers (2015) and Bestick et al. (2013) using linear programming approaches. Waugh et al. (2013)

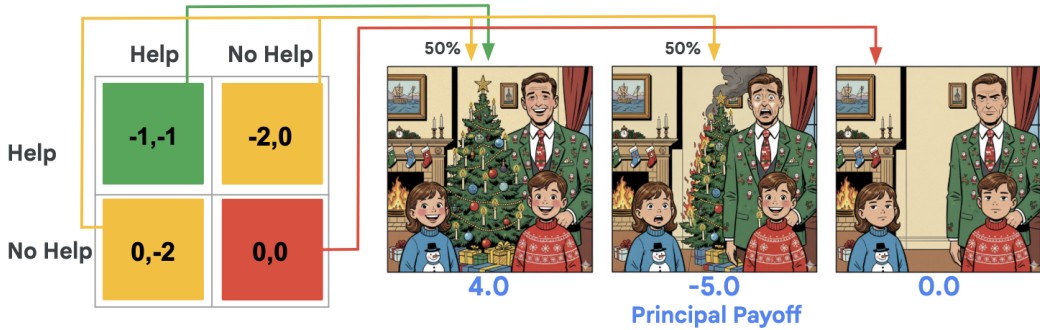

*Figure 2.* A simple example of a contract design problem with action space 2×2 and 3 outcomes.

*Table 1.* Experimental results. We report results for our approach (DID) and with and without local polishing (+Local) to give an approximate ceiling on the optimal local performance missed by DID. We also evaluate at the equilibrium computed by the DEB and ECOS to measure discrepancies from the DEB.

*(a)* Contract Design. Average change in principal utility from the generated contracts and the percentage non-harmful interventions ($\geq -10^{-4}$) in parentheses.

| EQL | DID (DEB) | +LOCAL (DEB) | DID (ECOS) | +LOCAL (ECOS) |
|---|---|---|---|---|
| CE | 0.25 (79%) | 0.51 (97%) | 0.05 (52%) | 0.29 (93%) |
| CCE | 0.33 (83%) | 0.59 (98%) | 0.20 (75%) | 0.52 (97%) |

*(b)* Inverse Equilibrium. Average KL distance between the generated payoffs and the target joint. We include a naive baseline (Uni) of the KL distance between the uniform joint to the target joint to ground the numbers.

| EQL | DID (DEB) | +LOCAL (DEB) | DID (ECOS) | +LOCAL (ECOS) | UNI |
|---|---|---|---|---|---|
| CE | 0.056 | 0.024 | 0.175 | 0.133 | 0.400 |
| CCE | 0.043 | 0.013 | 0.071 | 0.028 | 0.400 |

*(c)* Machine Scheduling. Average change in expected makespan (lower is better) from the generated taxes and the percentage of non-harmful interventions ($\leq 10^{-4}$) in parenthesis.

| EQL | DID (DEB) | +LOCAL (DEB) | DID (ECOS) | +LOCAL (ECOS) |
|---|---|---|---|---|
| CE | -0.339 (88%) | -0.408 (100%) | -0.096 (57%) | -0.352 (75%) |
| CCE | -0.211 (99%) | -0.240 (100%) | -0.050 (84%) | -0.077 (97%) |

focused on learning utility functions from stochastic observations of equilibrium behavior. A similar approach was taken by Syrgkanis et al. (2017) to learn the bidders' values from observing bids in different auctions.

Here we are interested in the following variant of the problem. Given a target equilibrium $\sigma^\omega$, which defines the context $\omega$, generate a game $G(\theta; \omega)$, such that its $\varepsilon$-ME-Eql minimizes the KL divergence to $\sigma^\omega$. Such problems that aim to compute game-theoretic equilibria, which are anchored to learned policies from (potentially human) expert

behavior are critical for many diverse ML applications as e.g. they can be used to discover near-optimal game play that is also interpretable and natural (Bakhtin et al., 2022; Jacob et al., 2022; Lerer & Peysakhovich, 2019; McIlroy-Young et al., 2020; Munos et al., 2024). This problem can be formulated as an instance of ($\varepsilon$-ME-ID) as follows:

$$\min_\theta \mathbb{E}_{\omega \sim \Omega} \left[ \text{KL}(\sigma^* \| \sigma^\omega) \right] \quad \text{s.t. } \sigma^* = \varepsilon\text{-ME-Eql}(G(\theta; \omega)).$$

To train the mechanism generator for this task we sample the joint action space $\mathcal{A}$ uniformly from $\{2\times2, \ldots, 16\times16\}$ and joint targets $\sigma_\omega$ uniformly from the simplex $\Delta^{|\mathcal{A}|-1}$. The loss computed is $KL(\sigma^* \| \sigma_\omega) + \lambda \| G(\theta; \omega) - \Pi(G(\theta; \omega)) \|^2$. Here $\Pi(G)$ is the equilibrium-invariant embedding of $G$ (Marris et al., 2023). This means $\| \Pi(G) \|_2 = 1$ and $\sum_{a_p} \Pi(G)_p(a_p, a_{-p}) = 0$. Note that both of these operations do not change the equilibrium set as long as the target $\varepsilon$ is scaled by the same factor as the payoffs. The penalty term thus only serves to enforce a standardized output format by decreasing the degrees of freedom. As it evaluates to zero on a subset of the possible outputs with minimum KL divergence it should not compromise the performance of the neural networks.

We evaluate on 8 samples for each game with shape in $\{2\times2, ..., 16\times16\}$ and report the average over all contexts. Due to the increased dimensionality of the search space, we polish using 4,000 function evaluations. The results can be found in Table 1b. The average KL divergence of our approach significantly outperforms the naive benchmark of outputting a constant game with the uniform joint as $\varepsilon$-ME-Eql. As before, there is some room for improvement if we polish the solution using a local optimizer. Unsurprisingly, we also see reduced performance when evaluating under ECOS, highlighting the approximate nature of the DEB. Note that for 2×2 games, we can identify the joint strategy space with $\Delta^3$ and thus plot our results on the simplex. Figure 3 illustrates two sampled targets $\sigma^w$, the $\varepsilon$-Eql polytope of the generated game $G(\theta, \omega)$ and the resulting $\varepsilon$-ME-Eql $\sigma^*$, which nicely matches the target.

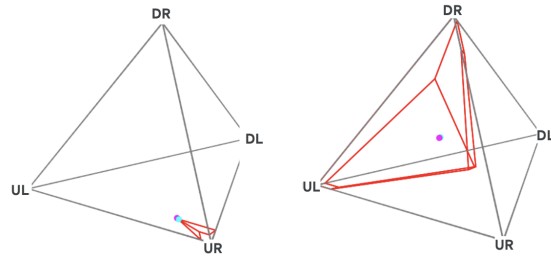

*Figure 3.* $\sigma^\omega$ is plotted in cyan, the 0.01-ME-CCE in magenta. The red polytope corresponds to the 0.01-CCE set. The underlying game is between a row player with actions **U**p and **D**own and a column player with actions **L**eft and **R**ight. The four vertices represent the four deterministic joint actions. Figure was produced with a smaller network trained on $2\times2$ games.

## 5.5. Machine Scheduling

The game-theoretic foundations of scheduling jobs to machines have been studied by Heydenreich et al. (2007) and Immorlica et al. (2009). The problem captures many real-world applications, such as fair load balancing of users to wireless access points (Bejerano et al., 2004), optimizing performance for content multihoming (Liu et al., 2012), and multiprocessor scheduling (Schuurman & Vredeveld, 2001). More recently, it has been revisited in the context of routing prompts to different LLM models (Cao et al., 2025; Hu et al., 2024; Jitkrittum et al., 2025; Ong et al., 2025).

Here we study the following variant. There are $N$ players and $M$ machines. Each player $p$ has a job and as an action $a_p \in \{1, \ldots, M\}$ can choose which machine to submit the job to. Each job-machine pair has an associated completion time $t_{p,j}$. If multiple players schedule to the same machine, the jobs get resolved in random order. The expected utility $u_p(a)$ is given by the negative expected completion time, $-t_{p,a_p} - \frac{1}{2}\sum_{p'\neq p, a_{p'}=a_p} t_{p',a_p}$. The context $\omega$ is thus equal to the sampled job times $t$ and we again use the superscript $t^\omega$ to refer to a specific draw.

We consider an incentive designer that wants to minimize the maximum makespan of the machines given by $\max_j \sum_{p:a_p=j} t_{p,j}^\omega$. We assume the designer can levy taxes $s_p(a; \theta, \omega) \geq 0$ on the agents, i.e. positive transfers from the agents to the designer, but wants to keep these minimal. The designer thus induces a game $G(\theta, \omega)$ with payoffs $G_p(\theta; \omega) = u_p(a; \omega) - s_p(a; \theta)$. Formulating the machine scheduling problem as an instance of ($\varepsilon$-ME-ID) yields

$$\min_\theta \mathbb{E}_{\omega \sim \Omega}\left[\sum_a \sigma^*(a) \max_j \sum_{p:a_p=j} t_{p,j}^\omega - \lambda\|s(\theta; \omega)\|^2\right]$$

$$\text{s.t. } \sigma^* = \varepsilon\text{-ME-Eql}(G(\theta; \omega)),$$

where the last term in the upper level serves as regularization to limit the tax magnitude.

In our experiments, we sampled the job times $t_{p,j} \sim$

Lognormal $(0, 0.5)$ and the number of machines uniformly from $\{2, \ldots, 12\}$ resulting in games of sizes between $2\times2$ and $12\times12$. To train our mechanism generator, we used as input the utility matrix $[u_1(\cdot; \omega), \ldots, u_p(\cdot; \omega)]$ with batch size 64 instead of the sampled job times. The reason is the utilities also contain the necessary contextual information but are of the same shape as the outputted taxes and thus more compatible with our equivariant architecture. To evaluate our algorithm we measure the change in makespan

$$E_{\omega \sim \Omega}\left[\sum_a \sigma^*(a) \max_j \sum_{p:a_p=j} t_{p,j} - \sum_a \sigma^0(a) \max_j \sum_{p:a_p=j} t_{p,j}\right],$$

where $\sigma^0$ is the $\varepsilon$-ME-Eql of the original game with $s_p(a; \theta, \omega) = 0$. The results can be found in Table 1c. We see that our mechanism generator, for the vast majority of sampled contexts outputs taxes, which decrease the makespan and that the decrease outperforms the benchmark in expectation. The small room for improvement when polishing (4,000 function evaluations) shows our method performs well. However, as before, evaluating under the ECOS distribution reduces performance.

## 6. Conclusion

In this work, we introduced *deep incentive design* (DID). It is a framework for solving mathematical programs with equilibrium constraints (MPECs), based on backpropagating gradients through differentiable equilibrium blocks (DEBs). By exploiting the differentiability and convexity of the $\varepsilon$-(coarse) correlated equilibrium polytope, we can train networks to predict the unique $\varepsilon$-maximum-entropy equilibrium of normal form games up to large scales and easily compute the gradients of these equilibria. Thereby, we unlock the whole toolkit of machine learning and gradient-based optimization to tackle the large class of relevant and hard game-theoretic incentive design problems.

Section 5 illustrated both the success of our approach on three diverse tasks and more importantly how easily it can be adapted to different problems. Indeed, with one single DEB (per desired solution concept), one fundamental type of equivariant layer, and a unified training pipeline with many fixed hyperparameters, we were able to train mechanism generators for three different problems, which themselves generalized across many game sizes and contexts. Given this generalizability, we anticipate that DID will be useful across the multi-agent community in tackling a wide array of problems with equilibrium constraints.

DID is a versatile framework that does not rely on specific solution concepts or architectures and opens several promising future research directions to complement our work. One example is extending DID to different equivariant architectures, such as transformers, or running it while continuously

training the DEB to avoid adversarial examples. Another is improving the scalability. The computation currently scales linearly with the game size $\prod_{p=1}^{N} |\mathcal{A}_p|$, which limits the number of feasible players. Therefore, studying our approach with succinct game representations, such as polymatrix games, could help scale to even larger strategic interactions. We are confident DID can be extended to such games with minor architectural adaptations. A complementary approach would be to combine DID with existing game abstraction techniques to tackle large-scale games (Sandholm, 2015; Zhang & Parkes, 2008; Li et al., 2024a). Another important research avenue is incentive design with equilibrium constraints. Hard constraints on fairness or welfare for example are particularly important if DID is to be implemented in the real world and—in cases where they are convex—should be straightforward to add to our framework. In general, we look forward to future applications of DID to other diverse problems with setting-specific DEBs, architectures and constraints.

## Acknowledgements

We would like to thank Ian Gemp, Siqi Liu, Paul Dütting, and Marc Lanctot for their constructive feedback and the helpful discussions during the writing of this paper.

## Impact Statement

This paper presents work whose goal is to use machine learning for the automated design of incentive systems that may serve many human users and become part of our economic systems. Machine learning algorithms are potent optimizers. This creates the risk of reward hacking, resulting from discrepancies between the specified proxy loss $\mathcal{L}$ and the designer's true intent. Furthermore, if the designer's objective is misaligned with user welfare, incentive design could result in a reduction of the latter. Practical deployments should therefore prioritize alignment and fairness constraints to prevent unintended consequences. Our framework can easily incorporate such constraints, if they are differentiable and convex.

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

# A. Experimental Details

## A.1. Architectures and Code

Our networks are built to respect the *equivariances* of the underlying game-theoretic functions. A function $f$ is said to be equivariant if applying a permutation $\tau$ to its input is equivalent to applying the same permutation to the output, i.e.

$$f(\tau(x)) = \tau(f(x)).$$

The game-theoretic functions we are trying to learn have several such equivariances, e.g. if we permute the action space of a player $p$ in a game $G$ to yield a game $G_\tau$ and do the same for $G$'s equilibrium $\sigma^*$ to yield $\sigma_\tau^*$ then $\sigma_\tau^* \in \text{Eql}(G_\tau)$ and we say the equilibrium map is equivariant with respect to the permutation $\tau$.

We briefly explain the fundamental architecture used for all layers in our network. All our data has a channel dimension and a varying amount of equivariant dimensions (such as players, strategies and outcomes). For a given input at channel $c$, $(c, \dots)$ we apply a set of *pooling* functions $\phi_i$ which preserve some corresponding equivariances $\tau_i$ of the equivariant dimensions, i.e.

$$\phi_i((c, \tau_i(\dots))) = (c, \phi_i(\tau(\dots))) = (c, \tau(\phi_i(\dots))).$$

The output from the pooling functions is then concatenated along the channel dimension and passed through a linear transform, whose weights are shared across the equivariant dimensions, i.e. only depend on the channel dimension. This ensures the equivariance is preserved and gives the network the inductive bias. Afterwards we apply a (nonlinear) activation function. In general we can thus recursively define the output $g_{l+1}$ of the $l+1$ layer as

$$g_{l+1}(c_{l+1}, \dots) = f\left(\sum_{c_l} \sum_{i=1}^{I} w(c_{l+1}, c_l, i)\phi_i(g_l(c_l, \dots)) + b(c_{l+1})\right),$$

where $I$ represents the number of different pooling functions applied, $C_l$ is the numbers of channels set for layer $l$ and $w$ is the weight of the linear layer. In all our experiments all hidden layers use GELU as the nonlinear activation function. For the final layers, the activation depends on the setting, which we will outline below.

All our code is written in Python. The neural networks are built using JAX (Bradbury et al., 2018) and Flax (Heek et al., 2024).

## A.2. Inverse Equilibrium Problem

### A.2.1. ARCHITECTURE

For the inverse equilibrium problem, the mechanism generator outputs payoffs of shape $[B, N, \mathcal{A}_1, \dots, \mathcal{A}_N, 1]$. It takes as input the target equilibrium $\sigma^*$ of shape $[B, \mathcal{A}_1, \dots, \mathcal{A}_N]$. Note, that there are many possible games that might have the same equilibrium. We are thus trying to learn a one-to-many mapping. To avoid the network averaging over the multimodal loss landscape, we also add a noise vector to the input. To match the output shape, we further expand and broadcast the input to shape $[B, N, \mathcal{A}_1, \dots, \mathcal{A}_N, 2]$, where the target joint is in the first channel and the noise is in the second channel. As the input and output are of the same shape, we will only use one type of equivariant layer in the architecture, called *P2P layer* (for payoff shape to payoff shape), which preserves the payoff shape and its natural equivariances. There are two different equivariances for the P2P layer to respect:

- For each player $p$, and permutation $\tau_p(1), \dots, \tau_p(|\mathcal{A}_p|)$ of their strategy space, it has to hold for all $c_l \in C_l, n \in N, (a_1, \dots, a_N) \in \mathcal{A}$ that $g_l(c_l, n, a_1, \dots, \tau_p(a_p), \dots, a_N) = \tau_p(g_l(c_l, n, a_1, \dots, \tau_p(a_p), \dots, a_N))$.

- For each permutation $\tau_N(1), \dots, \tau_N(N)$ of the players it has to hold for all $c_l \in C_l, n \in N, (a_1, \dots, a_N) \in \mathcal{A}$ that $g_l(c_l, n, a_{\tau_N(1)}, \dots, a_{\tau_N(N)}) = \tau_N(g_l(c_l, n, a_1, \dots, a_N))$.

In this work, we build a set of equivariant pooling functions $I$ that respects the above equivariance by combining a few fundamental equivariant operations. First, we identify a set of axes over which we can "reduce" the data, such that the equivariances are preserved. For the payoffs, these are (i) the player axis, and (ii) for a given player $p$ either their strategy axis $a_p$ or the strategy axes of all other players $a_{-p}$. For each possible axis, we can choose one of two possible equivariant

reductions: either the identity, which takes the current element, or the whole axis, which returns all elements in the axis. In a last step we apply an aggregation function over the returned elements from the axes. In our experiments we use the meanvar aggregation, which is defined as $\phi^{mv}(x) = \frac{\sum_{i=1}^{n} x_i}{\sqrt{n}}$ for $x \in \mathbb{R}^n$. However, other pooling functions, such as mean, sum, max, min etc. are possible.

With these operations we can sequentially define our equivariant pooling functions. First, we choose either the strategy axis $a_p$ or $a_{-p}$ then choose the reductions over the player and strategy axis and then choose one of multiple aggregation functions. In general, for $x$ axes, $y$ reductions and $z$ aggregation functions, this will yield $I = zy^x$ channels as input to the linear layer for each channel in the input of the equivariant layer. In our case $I = 2^3 = 8$. The 8 channels are made up of the 6 distinct pooling functions $\phi^{mv}, \phi_p^{mv}, \phi_{a_p}^{mv}, \phi_{p,a_p}^{mv}, \phi_{a_{-p}}^{mv}, \phi_{p,a_{-p}}^{mv}$, where the lower case represents the axes over which we apply the meanvar reduction and the first two functions are duplicated. Note that when we apply the meanvar reduction across all elements of a given axis, we broadcast the result across the axis to preserve the shape (and the equivariance).

In our experiments we use three hidden layers with 64 nodes each, i.e. $C_l = 64$. Note that after the pooling we have $C_l I = 512$ channels, which then get mapped back to $C_{l+1} = 64$ channels by the linear layer. The exception is the first layer, which only has 2 input channels and the final layer, whose output has channel dimension 1. Moreover, for the final layer we do not apply a GELU activation but the identity to ensure the network can generate both positive and negative payoffs.

### A.2.2. TRAINING PIPELINE

The training pipeline for the inverse equilibrium problem is illustrated in Figure 4

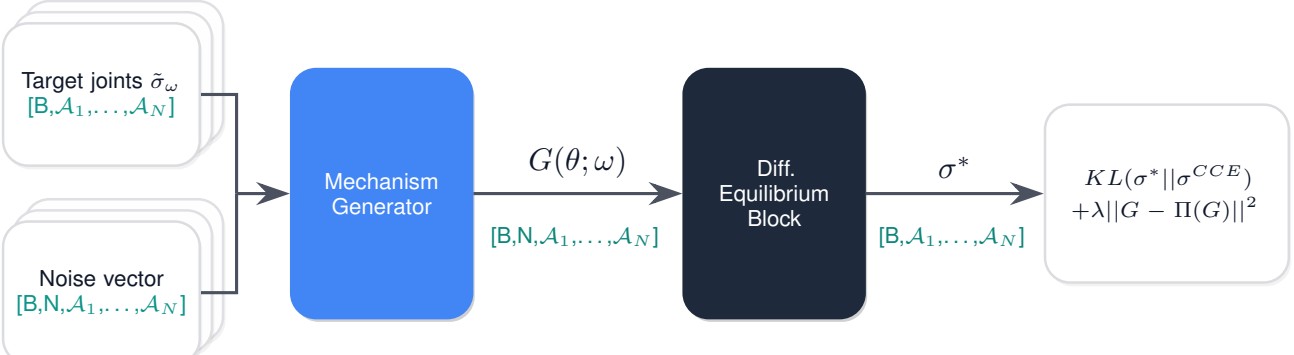

*Figure 4.* Inverse equilibrium training pipeline.

The target joints are sampled uniformly from the simplex and the noise from a standard normal distribution. The batch size is 64. The network is set up to expect an input shape of $[B, 2, 16, 16, 2]$. However by sampling masks uniformly at random, we train the generator to solve the task for games with sizes from $2\times2$ up to $16\times16$. We run the training with three seeds for $10^6$ steps using an exponential optimizer schedule with an ADAM optimizer (Kingma & Ba, 2015) and report the best run. The optimization parameters are $b_1 = 0.9, b_2 = 0.999$ with learning rate 0.01. We first increase learning rate to 0.01 over 50000 steps and then start ramping it down over 900000 steps. For each seed training takes about six hours on an H100.

### A.3. Contract Design

#### A.3.1. ARCHITECTURE

The architecture of the mechanism generator for contract design is illustrated in Figure 5.

For the contract design task, we have as input:

- The principal payoff of shape $[B, O]$.

- The samples payoffs of the agents of shape $[B, N, \mathcal{A}_1, \ldots, \mathcal{A}_N]$

- The sampled transition probabilities of shape $[B, \mathcal{A}_1, \ldots, \mathcal{A}_N, O]$.

In order to easily pass this as batched input through a neural network, we broadcast each to have a shape of $[B, N, \mathcal{A}_1, \ldots, \mathcal{A}_N, O]$ and concatenate them along a channel dimension.

The generator needs to compute contracts of shape $[B, N, O]$. These contracts are then transformed into expected payoffs of the agents using the transition probabilities and added to the initial payoffs before being handed to the DEB.

The change between input shape $[B, N, \mathcal{A}_1, \ldots, \mathcal{A}_N, O, 3]$ and output shape $[B, N, O]$, requires us to use three different layers:

1. First we pass the input through *PO2PO* layers. These work like the P2P layers described for the inverse equilibrium problem, using the same meanvar aggregation functions. However, they work with and preserve the shapes $[B, N, \mathcal{A}_1, \ldots, \mathcal{A}_N, O, C_l]$ and additionally preserve the equivariance with respect to permutations along the outcome axis. This means there is an additional axis, along which we can choose to reduce across all elements or just return the identity. Therefore, for every channel in the input, there are $2^4$ extra channels to map through the linear layer instead of $2^3$ for the P2P. To add to that our inputs have the extra output dimension, increasing the input size by a factor of $O$. This can lead to larger memory requirements and slow down training. To combat this, we split the PO2PO layer into three equivariant layers. First we reduce over the strategy axes leading to $4C_{l-1}$ channels and pass through a linear layer to reduce the channel dimension to $C_l$. Next we pool the player dimension leading to $2C_l$ channels, followed by another linear layer reducing the channel dimension to $C_l$ and finally pool over the outcomes to get $2C_l$ channels which get mapped to $C_l$ by another linear layer and a single nonlinear activation function. Training these three linear layers instead of one large linear layer roughly halves the number of weights we need to train.

2. Next, the *PO2O* layer maps the shape from $[B, N, \mathcal{A}_1, \ldots, \mathcal{A}_N, O, C]$ to the shape $[B, N, O, C]$ by applying the described pooling functions to the player and outcome axis and collapsing the strategy axes by applying the meanvar-pooling over all the points without broadcasting the result back to the original shape.

3. Finally, we have multiple *O2O* layers which take as input shapes $[B, N, O, C]$, apply the pooling functions along the player and outcome axis and then broadcast back to shape $[B, N, O, C]$ and concatenate along the channel. For the final layer, we set the channel dimension to be 1 and squeeze it. We also include a softplus activation to ensure positive contracts.

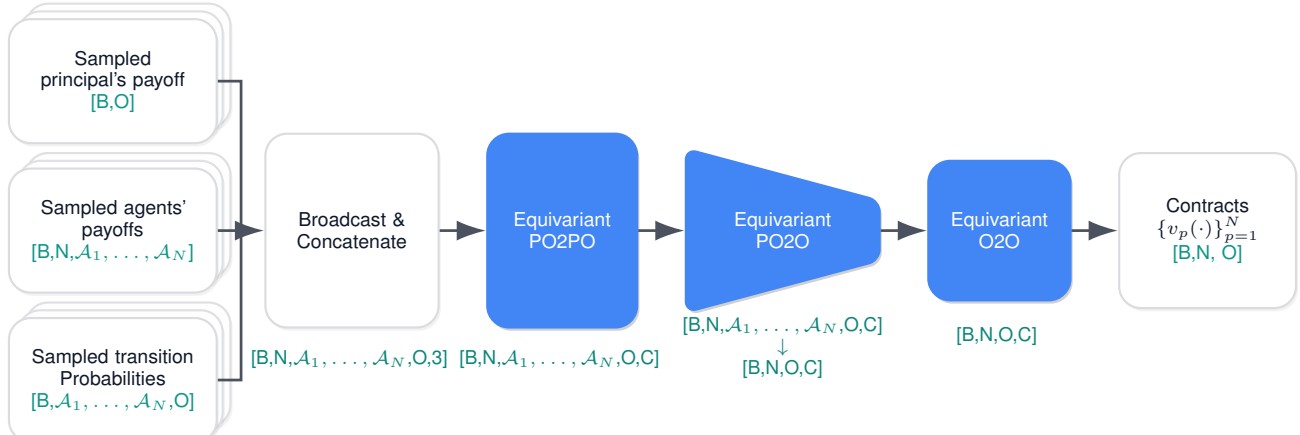

*Figure 5.* The architecture of the contract design mechanism generator. Blue boxes show the core generator and the green text shows the shapes.

### A.3.2. TRAINING PIPELINE

The training pipeline is illustrated in Figure 6. The network has three PO2PO layers with $C_l = 32$, one PO2O layer and two O2O layers with $C_l = 32$ nodes each. The batch size $B$ is 64 and we run the training for $2 * 10^6$ steps. We run it with three seeds each for both the 0.01-ME-CE and 0.01-ME-CCE and take the best run. The optimization is done with an exponential optimization schedule based on an ADAM optimizer (Kingma & Ba, 2015). The optimization parameters are $b_1 = 0.9, b_2 = 0.999$ with learning rate 0.01. We first increase learning rate to 0.01 over 50000 steps and then start

ramping it down over 900000 steps. Moreover, over 500000 steps we increase the penalty weight of the paid contracts (i.e. $\sum_p v_p(o; \theta, \omega)$) from 0 to 1 to avoid the network learning zero contracts early on. Each run takes about 15 hours on a NVIDIA A100 GPU. The batched input is set up to be of shape $[B, 2, 16, 16, 10, 3]$, so the network can handle up to 10 outcomes and $16 \times 16$ action space. However, we sample masks uniformly at random, which are also batched and passed to the network, so that within any batch we train the network for a broad distribution of outcomes between 2 and 10 and action spaces between 2 and 16 actions. The transitions are sampled from a Dirichlet distribution with $\alpha = 0.1$. The costs of the agents are sampled from $[-4, 0]$ and the principal payoff is uniformly sampled from $[-5, 5]$.

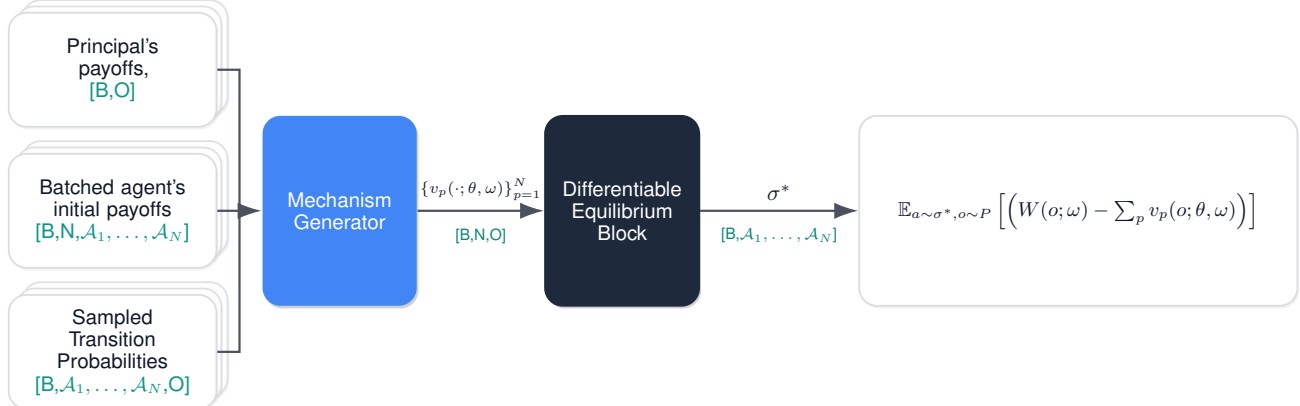

*Figure 6.* Training pipeline for the contract design task.

### A.3.3. ROBUSTNESS TO OTHER EQUILIBRIA

In contract design, where the designer has limited observability and control over the lower-level game, the question arises, what happens if the players in the lower-level game do not play the $\varepsilon$-ME-Eql, but instead converge to another equilibrium in the convex equilibrium polytope. First, note that the $\varepsilon$-ME-Eql corresponds to the point of the $\varepsilon$-Eql polytope closest to the center of the probability simplex. The principal's utility is a linear function and thus maximized, respectively minimzed at the vertices of the probability simplex. The $\varepsilon$-ME-Eql is thus never the joint strategy, which maximizes or minimzes the principal welfare over the whole simplex.

We did some empirical analysis of the shape of the equilibrium polytope for $2 \times 2$ games and found that when the contracts change the induced equilibrium towards the optimal joint action for principal welfare, the resulting equilibrium polytope is concentrated around this single vertex, making the $\varepsilon$-Eql the most pessimistic estimate over the equilibrium polytope with respect to the upper-level objective. Simultaneously, if the produced contracts do not change the induced game, the equilibrium polytope can be concentrated around the welfare-minimal vertex, in which case the $\varepsilon$-ME-Eql presents the most optimistic estimate of principal welfare under equilibrium play. Therefore, DID seems to be more robust the more successful the intervention was. However, further research on the shape of the equilibrium polytopes of the induced games, in particular in higher dimensions, is thus needed to better understand the robustness of DID with respect to varying equilibrium selection in settings where this can be a concern such as contract design.

### A.4. Machine Scheduling

#### A.4.1. ARCHITECTURE

We take as input the agents' payoffs of shape $[B, N, \mathcal{A}_1, \ldots, \mathcal{A}_N, 1]$ for the machine scheduling problem and output the induced payoffs of the same shape $[B, N, \mathcal{A}_1, \ldots, \mathcal{A}_N, 1]$. More precisely the generator learns taxes on each joint action of shape $[B, N, \mathcal{A}_1, \ldots, \mathcal{A}_N, 1]$ which are subtracted from the original payoffs. Therefore the equivariance and the network architecture is identical (3 hidden layers with size (64,64,64)) to the one used for the inverse equilibrium problem with the only difference being the final activation function, which in this case is $-\text{Softplus}(x)$ to ensure the change to the agents payoff is negative, as they are getting taxed.

#### A.4.2. TRAINING PIPELINE

The training pipeline for the machine scheduling problem is illustrated in Figure 7

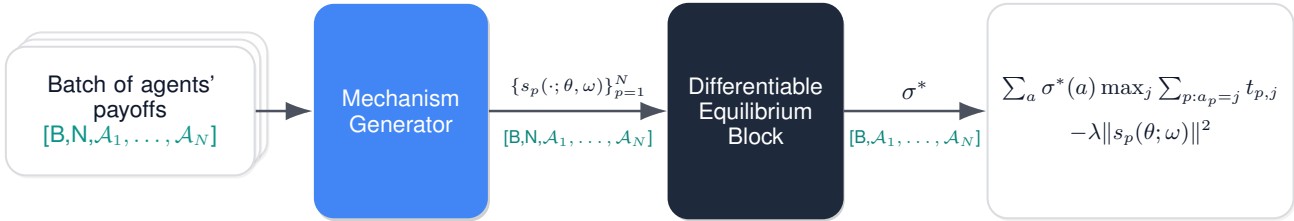

*Figure 7.* Training pipeline for the machine scheduling problem.

We sample the job time distribution $t_{p,j} \sim \text{Lognormal}(0, 0.5)$. From these we calculate the expected payoffs of the players using the formulas provided in Section 5.5. The batch size is 64 and we run three seeds and take the best run. The weights are optimized with an exponential optimization schedule based using ADAM (Kingma & Ba, 2015). The optimization parameters are $b_1 = 0.9, b_2 = 0.999$ with learning rate 0.01. We first increase learning rate to 0.01 over 50000 steps and then start ramping it down over 900000 steps. We also ramp up the penalty parameter $\lambda$ from 0 to 0.1 over 50000 steps. Each complete training run consists of $10^6$ training steps and takes roughly 10 hours on a NVIDIA A100 GPU.

### A.5. Differentiable Equilibrium Blocks

For all mechanism generators the final output is a batch of payoffs of shape $[B, N, \mathcal{A}_1, \ldots, \mathcal{A}_N]$, which gets passed to the differentiable equilibrium block.

We defer the reader to Marris et al. (2022) for a discussion on the architecture used for the DEB. The DEB we train consists of three layers of P2P layers with $C_l = 64$, one P2D layer and another three D2D layers with 64 nodes. Moreover, similar to the way we split the PO2PO layers into three linear layers for contract design, we split the P2P layer into two layers (one for reduction along the player and one for the reductions along the strategy axes) to save memory.

The DEB is set up to handle payoffs of shape $[B, 2, 16, 16, C]$. However, we implemented all layers to handle masked input. For each batch, we uniformly sample masks of sizes 0 to 30 to apply to the action space of both players, so that in each batch of size 64, we train on all games from sizes $2 \times 2$ up to $16 \times 16$. Across games, we empirically observe an $\varepsilon$ of 0.0198 for the CCE DEB and 0.039 for the CE DEB.

The games are sampled from the equilibrium invariant embedding (Marris et al., 2023) with their $L_2$ norm set to $\sqrt{\prod_{p=1}^{N} |\mathcal{A}_p|}$, where $|\mathcal{A}_p|$ is the effective action size after masking, to ensure unit variance of all inputs.

The loss function, as derived in Marris et al. (2022) is given by the dual formulation as:

$$L^{(C)CE} = \ln \left( \sum_{a \in \mathcal{A}} \hat{\sigma}(a) \exp \left( l^{(C)CE}(a) \right) \right) + \sum_p \epsilon^+ \left\{ \sum_{a'_p, a''_p} \alpha_p^{CE}(a'_p, a''_p), \sum_{a'_p} \alpha_p^{CCE}(a'_p) \right\} - \rho \sum_p \epsilon_p.$$

The logits $l^{(C)CE}(a)$ are given by:

$$l^{(C)CE}(a) = -\sum_p \left\{ \sum_{a'_p, a''_p} \alpha_p^{CE}(a'_p, a''_p) A_p(a'_p, a''_p, a), \sum_{a'_p} \alpha_p^{CCE}(a'_p) A_p(a'_p, a) \right\},$$

and the primal joint distribution $\sigma^{(C)CE}(a)$ is computed analytically as:

$$\sigma^{(C)CE}(a) = \frac{\hat{\sigma}(a) \exp \left( l^{(C)CE}(a) \right)}{\sum_{a \in \mathcal{A}} \hat{\sigma}(a) \exp \left( l^{(C)CE}(a) \right)}.$$

Similarly, the primal approximation parameter $\epsilon_p$ for each player is recovered via:

$$\epsilon_p = (\epsilon - \epsilon^+) \exp \left( -\frac{1}{\rho} \left\{ \sum_{a'_p, a''_p} \alpha_p^{CE}(a'_p, a''_p), \sum_{a'_p} \alpha_p^{CCE}(a'_p) \right\} \right) + \epsilon^+$$

Moreover, the following notation is used:

- $\alpha_p^{CE}(a_p', a_p'') \geq 0$ and $\alpha_p^{CCE}(a_p') \geq 0$: The dual deviation gains corresponding to the CE and CCE constraints, which are the outputs of the neural network.

- $A_p^{CE}(a_p', a_p'', a)$ and $A_p^{CCE}(a_p', a)$: The deviation gains, i.e.

$$A_p^{CE}(a_p', a_p'', a) = \begin{cases} G_p(a_p', a_{-p}) - G_p(a_p'', a_{-p}) & \text{if } a_p = a_p'' \\ 0 & \text{otherwise} \end{cases} \tag{1}$$

$$A_p^{CCE}(a_p', a) = \sum_{a_p'' \in \mathcal{A}_p} A_p^{CE}(a_p', a_p'', a) = G_p(a_p', a_{-p}) - G_p(a) \tag{2}$$

- $\hat{\sigma}(a)$: The uniform target distribution.

- $\epsilon$: The target approximation parameter.

- $\epsilon^+$: A maximum approximation parameter constant.

- $\rho$: A hyperparameter that controls the balance of the optimization by weighting the approximation distance term.

## B. In-depth Contrast to Previous Lines of Work

### B.1. Discussion

To situate our contribution of making (ID) accessible as a machine learning problem, we contrast deep incentive design with the most relevant precursors: (i) gradient-based MPEC solvers, (ii) differentiable economics, and (iii) Neural Equilibrium Solvers.

**Gradient-Based MPEC**  While gradient-based approaches have been tried before in the setting of (ID), the existing works have two important restrictions. First, instead of using $\epsilon$-ME-Eql as solution concept, they generally use the Nash equilibrium and have to make restrictive assumptions on its uniqueness and the monotonicity of the game. Second, the gradient computation is very expensive. Some approaches build on implicit differentiation, which requires inverting a matrix that is quadratic in the game size (Li et al., 2023). The remaining ones use automatic differentiation which requires running many steps of an iterative solver to converge to an approximate equilibrium and then backpropagating through the whole computational graph. In contrast, deep incentive design computes approximate gradients in time linear in $\mathcal{O}(|\mathcal{A}|)$ by backpropagating through the learned DEB. Moreover, our approach naturally learns to generalize between different contexts, whereas the classical gradient-based MPEC solvers need to be (re)run for a specific context.

**Differentiable Economics**  is similar in spirit to our approach. It consists of running gradient descent on neural networks to design a game. However, the existing literature has focused on Bayesian games—the natural setting for mechanism design problems. In those games, players are reporting their private types and the famous *revelation principle* states that any mechanism implementing an outcome in dominant strategies has an outcome-equivalent counterpart where it is a dominant strategy equilibrium for all players to report their types truthfully. Therefore, it is natural to restrict to searching within the space of truthful mechanisms. As the equilibrium is already known for these mechanisms, this circumvents the problem of differentiating through the equilibrium of the resulting game. Our focus is on non-Bayesian normal-form games with discrete action spaces, where truthfulness is not a salient concept as the players do not have private information. Therefore the problem of differentiating through the equilibrium persists and is tackled using our DEBs. We thus consider the existing works in differentiable economics as complementary to deep incentive design. Moreover, in the differentiable economics literature the neural networks have generally been trained as the mechanism themselves, taking as input a distribution of bids and outputting the prices for each problem instance. The network is used to abstract over the exponential type space. However, each task requires a different network and training run. In contrast our networks are not mechanisms but "mechanism generators", taking as input the distribution of contexts $\omega \sim \Omega$ and outputting the induced mechanism $G(\theta; \omega)$, thus abstracting over the space of contexts. They are trained once upfront and can be evaluated quickly for a broad class of problems, instead of having to be retrained when facing a different problem instance.

**Neural Equilibrium Solvers** There has been a line of work on using machine learning for learning game-theoretic equilibria. (Duan et al., 2023a; Bai et al., 2020; Jin et al., 2021; Hartford et al., 2016; Feng et al., 2021; Ling et al., 2018; McKenzie et al., 2024; Liu et al., 2024). In this work, our DEBs build upon the work of Marris et al. (2022), who proposed using layers with equivariant pooling functions for learning the $\varepsilon$-ME-Eql of a normal form game. Our DEBs further extend the work of Marris et al. (2022). Among our contributions is the fact that we implement an equivariant masking, which allows us to successfully train on a distribution of games of sizes between $2 \times 2$ and $16 \times 16$, compared to Marris et al. (2022) who train a single network per game size. This is a massive improvement in generalizability compared to previous works. More importantly, our work uses DEBs as one of multiple building blocks to solve a very broad class of incentive design problems. Our framework is the first to successfully backpropagate through such equilibrium blocks and proves they can be trained to have informative gradients and be integrated in larger workflows.

