# OpenReview forum: "Deep Incentive Design with Differentiable Equilibrium Blocks"
_ICML.cc/2026/Conference — ICML 2026 regular_

### Official Review · Reviewer_hwwb · 2026-02-14

**Soundness:** 2
**Presentation:** 3
**Significance:** 4
**Originality:** 3
**Overall Recommendation:** 4
**Confidence:** 4

**Summary:**

This paper proposes deep incentive design (DID) to address a wide variety of incentive design problems. Such problems have following dynamics: 1) a game context $w$ is randomly drawn; 2) a designer designs game parameter $\theta$, forming the normal-form game $G(\theta,w)$; 3) players response to $G(\theta,w)$ and form (coarse) correlated equilibria (CCEs or CEs) $\sigma^\star(G(\theta,w))\in Eql(G(\theta,w))$. The designer would like to minimize its expected loss: $\mathbb{E} _{w}[L(\theta,w,\sigma^\star(G(\theta,w)))]$.

This paper firstly learns a differentiable equilibrium block (DEB) to approximate $\sigma^\star(G(\theta,w))\in Eql(G(\theta,w))$. Specifically, given a normal-form game $G$, $\sigma^\star(G)$ is trained to approximate the maximum-entropy $\varepsilon$-approximate equilibrium. This paper then fixes the DEB to learn the mechanism generator $G(\theta,w)$, using standard first-order optimization on loss minimization with gradient computed through backward propagation. Experiments show that the DID approach benefits the designer, compared with the default game $G(w)$ where the designer gives no intervention on the game (here "no intervention" relies on the specific context of the studied setting).

**Compliance With Llm Reviewing Policy:**

Affirmed.

**Final Justification:**

This paper studies an interesting topic and proposes a novel approach. I still feel that the local polishment is not an ideal baseline to solve exact solutions in small settings, and some attempts to design alternative (possibly suboptimal) baselines can further strengthen the empirical result. I maintain my initial score.

**Key Questions For Authors:**

1. What's the accuracy of the pre-trained DEB block on the games in the stage of training DEB? Or on the games output by the mechanism generator in the stage of training the mechanism generator?
2. Can authors provide intuitions about why local improvements improves a lot ($\approx$ 2x) in contract design and inverse equilibrium? It seems to me that as long as the evaluation is same with training (post-process with a DEB), the learning of the mechanism generator is exact with finding a "local optimum" through first-order optimization.
3. See my other comments on Weakness 4.

**Limitations:**

yes

**Strengths And Weaknesses:**

Strengths:

* Incentive design is a really important problem. While existing works such as tax design and differentiale economics also lie in incentive design, they focus on specific settings. This paper aims at building a fundamental solution for incentive design, which provides a good starting point for understanding the whole body of incentive design and shows its great significance.
* The presentation is generally easy-to-follow.
* The idea of learning a general solution for incentive design is original (as far as I can tell). The three experiments are interesting and important, and I'm glad to see that the paper incorporates three settings into one framework.

Weaknesses:

1. There are some presentation issues throughout the papers.

(1). In section 2.3: The AI economist is proposed by Zheng et al. (2022), instead of Curry et al. (2022).

(2). It seems to me that $N=2$ always holds in experiments, please clearly mention that in Section 5 (if it is indeed the case). I tend not to criticize a small $N$ too much, as the normal-form game itself requires a representation that is exponential on $N$.

(3). Section A.5 is not finished, please complete it.

(4). It seems to me that there lacks an $N$ between $B$ and $A _1$ around line 846.

(5). In line between 178-183 and footnote 3, the authors claim that "$\sigma^*(\theta)$ is differentiable to $\theta$ follows from Rademacher's theorem". This claim seems correct, but it is not self-contained enough. As a result, I'm unclear about how the authors derive it. I would appreciate it if the authors could provide more details.

2. Experiments show that when the approach is evaluated with exact equilibria (ECOS) compared with DEB that returns approximate equilibria, there has a sharp decrease in performance. It suggests that mechanism generator is likely to learn from inaccuracies in DEB, instead of true equilibrium gradient, making the proposed approach less attractive. I appreciate the authors acknowledging this limitation and hints future works for further improvement.
3. The proposed approach seems to scale poorly to large $N$ due to the curse of dimensionality. As a contrast, incentive design such as tax design and mechanism design usually scales polynomially with $N$.
4. I have some concerns that challenge the technical soundness of this paper:

(1). The only baseline is not to intervene the game for the designer (and a naive baseline in *Inverse Equilibrium*). In all experiments, there are no inter-correlated constraints with $w$. That means, by sampling $w$, one could get a two-stage game with one designer and multiple players. Then, one could design simple baselines, or incorporate existing approaches --- those mentioned in the first paragraphs of Section 5.3 and 5.4 --- into the studied settings. It seems to me that the authors should have considered more comprehensive baselines, unless these baselines are inappropriate to these settings.

(2). In the last paragraph of section 5.3, the paper claims "For both CEs and CCEs, the learned $\theta^\star$ produces contracts that significantly improve the principal’s utility over no intervention". It's currently unclear to me whether the improvement can be regarded as significant. For example, are contracts produced by $\theta^\star$ close to an optimal one?

---

> ### Author Rebuttal · Authors · 2026-03-31
>
> We thank the reviewer for their effort and detailed, helpful comments. We also appreciate your acknowledgement of the generality and significance of the DID framework and the breadth of our experiments. Below we address the points you raise.
>
> **Presentation/Typos** Thank you very much for the detailed reading and the helpful pointers. These are addressed for camera-ready.
>
> **Scalability compared to mechanism design/tax design** It is not true that “mechanism design usually scales polynomially with N”. In the general case, allocation and payment rules must be defined over the full type space, which is exponential in N (see e.g. Conitzer et al. Automated Mechanism Design for a Self-Interested Designer for a detailed discussion on this). Polynomial scaling in existing mechanism design work is mostly achieved by restricting to succinct representations (e.g., XOR bidding languages, single-parameter domains, etc.). The analogous path for DID is to exploit succinct game representations such as polymatrix or congestion games, where the (C)CE can be represented compactly. We discuss this in Section 6 and believe it is an interesting research direction to build upon our framework. In this initial work however, rather than restricting to a specific, structured subclass to obtain polynomial worst-case guarantees, our goal was to establish a general-purpose framework that covers the broadest possible problem class and to demonstrate that machine learning tools trained on this broad distribution can achieve strong performance in practice. We have added a brief discussion of this to the camera-ready version.
>
> **Experiments** We understand this concern and took it seriously during our experiments. We therefore want to emphasize three points:
>
> 1) Our formulation of incentive design is novel and there are thus no existing algorithms that could serve as a good baseline throughout the diverse tasks.
>
> 2) We also actively looked for setting-specific baselines. But most of the works on contract design that we mentioned in Section 5.3 make very restrictive assumptions (linear contracts, binary action spaces) and focus on approximation and complexity results and not on developing implementable algorithms. Similarly, the works on inverse equilibrium problems are not directly applicable.
>
> 3) We therefore thought about the naive baseline approach of e.g. performing grid search with a lower-level iterative equilibrium solver. However, this simply does not scale to the high dimensionality of the problem (up to 16x16 games). It is thus not an informative baseline and not included in our paper. We therefore opted for the “local polishing” approach that reliably acts as a lower bound on the gap between our solution and the (intractable to compute) global optimum.
>
> **Learning inaccuracies of the DEB** Thank you for acknowledging that we deal transparently with this limitation. Although even under exact ECOS evaluation, DID still yields meaningful improvements, a co-training setup between the DEB and the generator to prevent distribution shift/adversarial examples is a clear practical consideration for implementing DID in future work. Hence, why we raised this in Section 6.
>
> **Claim in Section 5.3.** It is computationally intractable to compare our results to the true global optimum. Nevertheless, our learned interventions produce a measurable, consistent positive effect over no intervention, particularly for CCEs. We can tone down our wording, e.g., replacing "significantly improve" with "consistently yield measurable improvements".
>
> **Accuracy of the DEB** Good question. During training, we set the target $\varepsilon=0.01$ for the DEB, which is needed to ensure differentiability (cf. our comment to reviewer gQnr). Averaged over all games and all game sizes, the mean gap of the trained DEB is about 0.0198 for CCE and 0.039 for CE. Those results are comparable to the ones observed by Marris et al., who introduced the Neural Equilibrium Solver we are using. We have added this information to the Appendix of the camera-ready version.
>
> **Local Polishing Improvements** This is a great question. The local polishing optimizes each context individually.  The mechanism generator, in contrast, is trained to minimize expected loss over the entire context distribution $\Omega$ with a fixed-size network. It thus necessarily makes compromises/has an abstraction error across contexts, even though it is trained on the same gradients/lower-level solver as the local polishing. Closing this gap with larger networks, better architectures etc. is an interesting research direction to explore.
>
> We hope this clarifies your questions and can help you in your final assessment of our paper. We are also happy to discuss further questions during the discussion phase.

---

> > ### Author Rebuttal · Reviewer_hwwb · 2026-04-03
> >
> > Thank you for the authors' response.
> >
> > While most of my initial concerns are addressed, I have a few follow-up questions to the authors.
> >
> > 1. The authors mentioned in the rebuttal, "In the general case, allocation and payment rules must be defined over the full type space, which is exponential in N". I would like to point out that some works (e.g., Dutting, P. (2024) as cited) use neural networks to represent the allocation and payment rules, which is polynomial-time.
> > Well, I understand that this work is an initial work, so it's okay to focus on normal-form game representation, although it brings exponential complexity to N. I tend not to criticize this point anymore.
> >
> > 2. For the choice of baseline, I believe that it's pretty important to compare the proposed framework to setting-specific baselines or optimal solutions in some sense. It will be acceptable even if results are reported only on small instances (e.g., 2x2 or 4x4). Without such comparison, it's currently hard to verify whether the proposed DID framework (as well as local polishing) is close to optimal. I believe that there must be some algorithms for general models of contract design and inverse equilibrium (Feel free to point out whether I'm incorrect).
> >
> > Given the currently unresolved concern about baselines, I maintain my initial score.

---

> > > ### Author Response · Authors · 2026-04-08
> > >
> > > We thank the reviewer for their thoughtful follow-up and answer your questions below.
> > >
> > > 1. This is a fair point. We agree Duetting et al.’s approach circumvents having to explicitly define the payment and allocation function over the exponential type space by using a neural network as function approximator.
> > > There is an interesting parallel to be drawn here: both DID and Dütting et al. use neural networks to abstract over one exponential axis via sampling. In their case, the network abstracts over the space of type profiles (saving the exponential over types). In our case, the mechanism generator abstracts over the space of contexts $\omega \sim \Omega$ (saving the exponential over problem instances, each of which would otherwise require solving a separate MPEC). Neither approach addresses the per-instance representation complexity: for Dütting et al., a single bidder's combinatorial valuation is $2^m$, which is why they restrict to small $m$; for DID, a single normal-form game has $\prod^n |A_p|$ entries, which limits the number of players we can handle. We will add this parallel to our existing comparison of DID and differentiable economics in the Appendix for camera-ready. Thank you for your help in developing this point.
> > >
> > >     Overall, we agree with you that (i) this is an initial work designed to capture the broadest class of problems (ii) it is an    interesting future direction to study subclasses that have polynomial representations.
> > >
> > > 2. We share your view on the importance of benchmarking. We want to make two points. First, we have actively searched for setting-specific baselines, including reaching out to researchers working on multi-agent contract design, and have not been able to identify existing implementable algorithms that apply to our setting. A key reason is that our use of the maximum-entropy (C)CE as the lower-level solution concept is itself novel, making direct comparison to prior methods difficult, as none are designed for this constraint.
> > >
> > >     Second, we want to highlight that for small games (e.g., 2×2 as you mention), our local polishing procedure, which uses Nelder-Mead initialized from the DID solution with thousands of function evaluations, is essentially an exhaustive local search over the parameter space. Given the low dimensionality of 2×2 games (4 payoff entries per player), this should be close to the global optimum, giving us confidence that the gap reported in Table 1 is a meaningful measure of DID's quality. Thank you for making this point. We will look into highlighting this in the paper.
> > >
> > > We hope this answers your follow-up questions. Once again we want to thank you for your time and effort in reviewing our paper and the detailed and helpful feedback.

---

### Official Review · Reviewer_gQnr · 2026-02-16

**Soundness:** 3
**Presentation:** 3
**Significance:** 4
**Originality:** 3
**Overall Recommendation:** 5
**Confidence:** 3

**Summary:**

This paper introduces Deep Incentive Design (DID), a framework for solving mathematical programs with equilibrium constraints (MPECs) by leveraging differentiable neural networks. The core idea is to replace the lower-level equilibrium selection with a pre-trained differentiable equilibrium block (DEB) that computes the ϵ-ME-(C)CE of a normal-form game. The upper-level mechanism generator is trained end-to-end via gradient descent, enabling scalable incentive design across multiple contexts. The method is evaluated on three tasks: multi-agent contract design, inverse equilibrium problems, and machine scheduling. Results demonstrate that DID can learn effective mechanisms and generalize across game sizes and contexts.

**Compliance With Llm Reviewing Policy:**

Affirmed.

**Final Justification:**

My concerns are solved. I improve my confidence accordingly.

**Key Questions For Authors:**

1. What is the total training time and computational budget for the DEB and the mechanism generator across all experiments?

2. How sensitive is the training to the choice of ϵ? Does a smaller ϵ make gradients noisier or harder to train? Have the authors experimented with different values?

**Limitations:**

yes

**Strengths And Weaknesses:**

Strengths:

1. The paper successfully reformulates a classic game-theoretic MPEC into a differentiable learning pipeline, opening the door for modern deep learning tools to tackle incentive design.

2. By using equivariant neural networks with pooling over players, actions, and outcomes, the approach achieves strong inductive bias, parameter efficiency, and generalization across game sizes.

3. The framework is demonstrated on three diverse and practically relevant problems, showcasing its flexibility and potential impact.

4. The work seems the first learning-based general mechanism design framework for general-sum multi-player NFGs.

5. The paper is well-motivated and clearly explained.

6. DID advances a unified, modular approach to a broad class of incentive design problems that often require bespoke optimization pipelines.

Weaknesses:

1. The DEB is pre-trained in a supervised manner on randomly sampled games and is not updated during mechanism generator training. This could lead to distribution shift: the generator may exploit inaccuracies in the DEB (as hinted in Table 1, where ECOS evaluation shows performance drops). In real-world deployment, where induced games may lie outside the DEB’s training distribution, this could cause significant degradation. A self-play or co-training approach (where the DEB is continuously updated) might improve robustness.

2. The method scales linearly with the full joint action space, which becomes prohibitive for games with many players or large action spaces. While the paper mentions this limitation, it does not propose or evaluate any mitigation (e.g., factorized game representations, sparse equilibria, or sampling-based methods). Incorporating learning-based abstraction techniques [1-4] into the DID pipeline could be a promising direction for scaling the framework to domains with inherently massive action spaces.

3. The paper mentions that fairness or welfare constraints could be added if differentiable and convex, but no experiments or concrete guidance are provided. In real-world incentive design (e.g., taxation, contracting), such constraints are often critical. The absence of this exploration weakens the claim of practical deployability.

4. The framework assumes differentiable and convex constraints, which may not hold for all real-world design objectives.

Minor: Crowded layout in Line 370; Incomplete Statement in Line 950.

[1] Boning Li, Zhixuan Fang and Longbo Huang. RL-CFR: Improving Action Abstraction for Imperfect Information Extensive-Form Games with Reinforcement Learning. ICML 2024

[2] Carlos Martin and Tuomas Sandholm. Joint-perturbation simultaneous pseudo-gradient. IJCAI 2025

[3] Boning Li and Longbo Huang. Efficient Online Pruning and Abstraction for Imperfect Information Extensive-Form Games. ICLR 2025

[4] Carlos Martin and Tuomas Sandholm. Solving Infinite-Player Games with Player-to-Strategy Networks. arxiv 2025

---

> ### Author Rebuttal · Authors · 2026-03-31
>
> We thank the reviewer for the thoughtful and detailed feedback and for highlighting DID as a novel, unified framework and recognizing our parameter-efficient architecture. We appreciate the “Accept” recommendation and give our responses below.
>
> **Scaling** Thank you for pointing this out. While DID is a big leap in terms of what we can usually solve with game-theoretic methods, we agree that combining it with abstraction techniques and sparse game representations (such as the polymatrix games we discussed in Section 6) could help us scale this approach further. We are very excited about this research direction and thank the reviewer for pointing us to some of the related literature. We will happily add a discussion on abstraction methods for scaling DID to the camera-ready version.
>
> **Constrained optimization** You raise a fair point. A full experimental exploration of constraints was beyond the scope of this initial work, where our aim was to lay the conceptual groundwork for DID. While we believe that several natural constraints are indeed convex (for example bounds on the interventions’ magnitude in contract design and machine scheduling), we acknowledge that handling non-differentiable or non-convex constraints remains a harder, open challenge. We have clarified this for camera-ready.
>
> **Total training time & Compute Budget** For the DEB of a specific equilibrium concept, the training is run on a single GPU (A100) and takes approximately 12 hours. Once the DEB is trained, each mechanism generator (for a specific experimental setting and equilibrium concept) takes roughly another 12 to 15 GPU hours. For the two equilibrium concepts and three experimental settings studied, this totals roughly 96 GPU hours. The different tasks and solution concepts are of course parallelizable so you can train everything in a single. Moreover, this can be sped up with HP tuning and fewer steps.
>
>  **Distribution Shift** We agree that distribution shift between the DEB's training distribution and the games produced by the mechanism generator is an important practical consideration. As discussed in Section 5, this is why we included the ECOS baseline to transparently quantify the effect. However, we want to highlight that even under ECOS, the generated games consistently improve over no intervention/the uniform baseline across all three tasks and all possible games with 2x2 up to 16x16 actions. This clearly suggests the gradients are informative, useful and flowing through the network. Moreover, we train on the whole distribution of games, such that it cannot be the case that “games may lie outside the DEB’s training distribution”. Having said that, we agree that a continual learning setup, where you continuously train the DEB on the distribution of games produced by the mechanism generator is one of the primary research directions to strengthen and improve those results in future work and thus discuss this in Section 6.
>
> **Sensitivity of the choice of $\varepsilon$** This is a great question. We require $\varepsilon >0$ to guarantee differentiability (cf Section 4.1). This creates a trade-off as larger $\varepsilon$ yield smoother gradients, but weaker equilibrium guarantees. During research, we therefore ran initial experiments with both $\varepsilon =0.01$ and $\varepsilon =0.1$. We found that $\varepsilon=0.01$ converged just as robustly but yielded superior downstream results (e.g., higher welfare in contract design), as the strategies remained more deterministic. We concluded $\varepsilon=0.01$ provides the right tradeoff of ensuring a smooth gradient flow and equilibrium strictness.
>
> **Minor typos** Thank you for reading the paper closely and pointing these out. These have been fixed for the camera-ready version.
>
> We hope this addresses your questions and increases your confidence in the framework and your final assessment. We are happy to answer any further questions during the discussion phase!

---

> > ### Author Rebuttal · Reviewer_gQnr · 2026-04-02
> >
> > Thank you for your thorough and thoughtful rebuttal. I appreciate that you have addressed my concerns. I am increasing my confidence.

---

### Official Review · Reviewer_V6ah · 2026-02-20

**Soundness:** 3
**Presentation:** 2
**Significance:** 2
**Originality:** 2
**Overall Recommendation:** 4
**Confidence:** 4

**Summary:**

This paper proposes a computational framework termed Deep Incentive Design for automated economic mechanism design in multi-agent settings using neural networks. The key technical contribution is the introduction of Differentiable Equilibrium Blocks, which leverage equivariant neural networks to compute $\epsilon$-maximum-entropy coarse correlated equilibria. By applying the implicit function theorem, the framework enables gradients to propagate smoothly through the equilibrium-solving module during backpropagation. The authors empirically evaluate DID on three discrete-game benchmarks: multi-agent contract design, inverse equilibrium inference, and machine scheduling.

**Compliance With Llm Reviewing Policy:**

Affirmed.

**Final Justification:**

I raise the score because the authors addressed most of my concerns. I hope the paper will incorporate broader evaluation and further experiments on scalability in future work.

**Key Questions For Authors:**

See above.

**Limitations:**

yes

**Strengths And Weaknesses:**

**Strengths:**
- Well-written and easy to follow.
- The paper provides a clean end-to-end formulation of a bilevel mechanism design problem using implicit differentiation. Treating the equilibrium solver as a differentiable module is a reasonable and practical design choice, and avoids the need to unroll the solver during backpropagation.
- The DEB architecture is thoughtfully implemented with equivariant layers and masking, allowing a single model to handle games with different action space sizes in the same batch. This makes the approach more flexible than a fixed-size or toy implementation.

**Weaknesses:**
- The experimental comparison is quite limited. Most baselines are either naive heuristics or exact solvers, and it is unclear how DID compares to existing learning-based mechanism design approaches such as RegretNet-style methods. How does DID perform relative to these prior methods in terms of solution quality and runtime?
- Despite claims of generality, the experiments are restricted to relatively small discrete normal-form games. It is unclear how the approach would scale to more realistic settings with continuous, high-dimensional, or Bayesian action spaces without running into discretization issues. Can the authors clarify how the framework might extend to such settings?
- Although implicit differentiation helps with backpropagation memory, the forward pass still requires solving an equilibrium inside the DEB module at every update. This could become computationally expensive as the game size grows, raising concerns about scalability to larger problems.
- The use of a maximum-entropy equilibrium simplifies gradient computation, but real agents may converge to different or even worst-case equilibria. The paper does not discuss how sensitive the learned mechanisms are to such equilibrium selection issues. What happens if agents deviate from the assumed equilibrium behavior?

---

> ### Author Rebuttal · Authors · 2026-03-31
>
> We thank the reviewer for their time, effort and comments. We appreciate the reviewer recognizing the clean end-to-end formulation of DID and the game-theoretically equivariant architecture. Below we address each of the points you raised.
>
> **Forward pass requires costly equilibrium solver**.   **_This is a misunderstanding that we wish to clarify._** The DEB is a pretrained neural network. We do *not* invoke any iterative solver during training or inference of either the DEB or the mechanism generator. The forward (and backward) pass are regular neural network calls and thus linear in the game size $|A|$. _This speedup we get from performing automatic differentiation through the DEB network–in contrast to implicit differentiation, or differentiating through an iterative solver–is one of the core strengths/contributions of our work._
>
> **Equilibrium Sensitivity** This is an interesting question on the robustness of DID with respect to the selected equilibrium. We make two points:
>
> First, the choice of max-entropy equilibrium is not a rigid assumption. DID works with other strongly convex selection functions over the (C)CE polytope that yield a unique, differentiable choice. Max-entropy is a natural default (well-studied etc.), but researchers can substitute an alternative selection function tailored to their application.
>
> Second, the sensitivity to equilibrium selection is application-dependent. In the inverse equilibrium problem, multiplicity is not a concern and thus not discussed. Having additional equilibria does not undermine the target joint being implemented as an equilibrium. In contract design sensitivity is more important, which is why we in fact discuss it in Footnote 8 (lines 324-329). If players converge to a different equilibrium it could adversely affect the principal’s welfare. We investigated this experimentally and found more often than not that the $\varepsilon$-ME-Eql represents a rather conservative estimate, as it lies in the interior of the polytope, far from the vertices where the linear welfare objective is maximized. For camera-ready, we expanded this discussion in the Appendix.
>
> **Scalability to continuous action spaces** Our generator can output 16x16 payoffs for each player. We are thus successfully optimizing over a 16x16x2=512-dimensional space. By game-theory standards, considering the complexity of equilibrium computation, those are very large games.
> That said, we agree it would be interesting to combine our methods with game abstraction techniques to tackle games with continuous action spaces but that bounding the abstraction error would require work. As discussed in Section 6, we thus think the most promising approach to scale up our technique is to consider sparse games (e.g. polymatrix games).
>
> **Comparison to prior methods** We do _not_ agree with your remark that “it is unclear how DID compares to [...] approaches such as RegretNet”. We clearly mention Dütting et al. (who introduced RegretNet) and related works in Section 2.2. We further contrast DID to differentiable economics (including RegretNet) in detail in Appendix B (lines 970-984).
>
> RegretNet and DID share the same philosophy of applying ML to game theory. However, the two lines of work solve fundamentally different problems and cannot be evaluated on the same benchmarks. In particular, RegretNet operates in Bayesian games, where players need to reveal their type. In this setting the revelation principles allows us to restrict to truthful mechanisms, meaning the equilibrium is *fixed by construction* (truthful reporting). In contrast, DID operates in non-Bayesian normal-form games without a notion of a canonical “truthful” equilibrium. The core technical challenge we solve–differentiating through the equilibrium map–does not even arise in the RegretNet setting.
>
> To add to that, RegretNet learns over the distribution of bids for one single auction and the weights of the network thus represent the mechanism for that specific problem. In contrast in our setting we learn a *mechanism generator* that is trained over the distribution of all possible problems in that class.
>
> Because the problem formulations, inputs and outputs differ, there is no shared benchmark on which both methods can be meaningfully compared. For a broader discussion on our choice of baselines, we refer to our rebuttal for reviewer vM2u.
>
> **Final Remark**
> We hope our answers addressed your concerns and clarified the misunderstanding on the equilibrium solver, as well as the relation to RegretNet approaches. We therefore would be grateful if you could reflect these in your final assessment of our paper and are happy to continue the discussion during the discussion phase.

---

> > ### Author Rebuttal · Reviewer_V6ah · 2026-04-02
> >
> > Thank you for your response.
> >
> > I understand the theoretical distinction between Bayesian (RegretNet) and non-Bayesian settings, and I accept that a direct comparison on the same benchmark is infeasible. However, introducing a new differentiable framework without comparison to any learning-based or heuristic baselines makes it difficult to assess the empirical advantage of DID.
> >
> > Follow-up Question About Scalability: Thank you for contextualizing the 16x16 dimensionality. Given that scaling to continuous spaces requires bounding abstraction errors, I would like to better understand the limits of the current DEB architecture. What are the concrete bottlenecks preventing training the DEB network for moderately larger discrete spaces (e.g., 50x50 or 100x100)? Is it the memory footprint, the sample complexity required for pretraining, or the representational capacity of the network?

---

> > > ### Author Response · Authors · 2026-04-08
> > >
> > > Thank you for your acknowledgement and interesting follow-up.
> > >
> > > To your point on baselines, we want to briefly note that the Nelder-Mead method acts as a heuristic baseline. In particular, for the smaller games with low dimensionality it should yield similar results to running a grid search or gradient descent, but will scale better to higher dimensions. We will try to clarify this in our description of the method for camera ready.
> > >
> > > To answer your follow-up question: Marris et al. (Turbocharging Solution Concepts: Solving NEs, CEs and CCEs with Neural Equilibrium Solvers) successfully trained the equivalent of our DEB up to 64x64 and later work by Liu et al. (NfgTransformer: Equivariant Representation Learning for Normal-form Games) also successfully trained a DEB of size 16x16x16. During the writing of this paper, we ran some settings up to 32x32, which worked very well e.g. for the inverse equilibrium task. We are thus confident that with additional hyperparameter tuning 64x64 is achievable. The main bottleneck we observed when scaling up was convergence to bad local minima, as the search space grows combinatorially. Because our equivariant architecture is agnostic to game size, this can be tackled either via a curriculum that progressively increases game size during training, or via the masking approach we already use, where each batch contains games of diverse sizes so the model learns useful inductive biases from smaller, easier instances.
> > >
> > > The memory footprint for the forward and backward pass is not a practical concern at current scales. Doubling the action space per player leads to a quadratic increase in memory, so this will definitely become the limiting factor, but we do not see it as a bottleneck for scaling to e.g. 64x64. In our experiments we still used a batch size of 64 so there was sufficient GPU memory left. Similarly for network capacity, our equivariant architecture is very parameter-efficient. In our preliminary 32×32 experiments we did not need to increase the network size (or decrease the batch size). Therefore, even if capacity becomes an issue at, say, 64×64, it should be addressable by moderately increasing width or depth (and in turn potentially decreasing batch size).
> > >
> > > We hope our reply answered your remaining question and gave you a better understanding of the scalability. If so, given the previously resolved misunderstanding on the efficiency of the DEB forward pass, we would be grateful if you could consider revising your score in your final assessment. In any case, we thank you for your effort and time in reviewing our work and the helpful feedback we received.

---

### Official Review · Reviewer_vM2u · 2026-03-10

**Soundness:** 3
**Presentation:** 3
**Significance:** 3
**Originality:** 3
**Overall Recommendation:** 4
**Confidence:** 3

**Summary:**

The paper proposes Deep Incentive Design (DID) for solving mechanism design problems with deep learning building blocks. The method composes a mechanism generator with a pretrained differentiable equilibrium solver block to optimize for a design objective. It approximates the gradients by differentiating maximum-entropy correlated/coarse correlated equilibria. Experiments demonstrate the approach on three principal-agent problems such as contract design, inverse equilibrium generation, and machine scheduling problems. The implementation uses game-theoretic equivariance to reduce the number of parameters (e.g., the game is equivariant over players or player's strategies). The empirical evaluation compares 1) the performance when equilibrium is computed by the blackbox DEB and by the solver; 2) the local improvement missed by DID (since global optimal is not tractable, the comparison initiates a solver at DID's solution).

**Compliance With Llm Reviewing Policy:**

Affirmed.

**Final Justification:**

I keep my weak acceptance assessment of the paper. The author's response clarified that the hardness of the problem itself limits the evaluation strategies. I fully understand this limitation, which is why I'm leaning towards acceptance.

**Key Questions For Authors:**

I wonder if it is possible to bound the performance in terms of the error of DEB system?

**Limitations:**

yes

**Strengths And Weaknesses:**

The paper explains the components of the method very clearly. The method is also very general. The central empirical weakness is the reliability of the learned gradients, according to the experiment results. It seems like performance may drop a lot when DEB is substituted by exact solvers, suggesting the generator may actually exploit the approximation nature of DEB. The only baseline is for the inversion equilibrium generation problem, where the constant baseline seems weak.

---

> ### Author Rebuttal · Authors · 2026-03-31
>
> ## Comment to all Reviewers
>
> We sincerely thank all reviewers for their constructive feedback and time. We are encouraged by the consensus that our work focuses on a significant issue in automated game design, and that DID “advances a unified, modular approach to a broad class of incentive design problems that often require bespoke optimization pipelines”. Moreover, we appreciate the reviewers’ recognition of our equivariant, parameter-efficient implementation of the DEB and generator architecture allowing us to handle different sized games in the same batch. We address the technical questions and points raised individually below.
>
> ## Comment to Reviewer vM2u
> We thank the reviewer for their time and effort. We appreciate that you liked the clarity of our paper and the generality of our proposed framework.
> We address your questions and points below:
>
> **DEB-to-ECOS gap** We emphasize that even under ECOS, the generated interventions still consistently improve over no intervention. This is a strong result, given we evaluate over *all* possible games with 2x2 to 16x16 actions. Having said that, we agree this is a practical limitation of the algorithm in its current form, which is why we included the ECOS baseline to quantify the effect and transparently discussed the issue in Section 5. As mentioned in Section 6 of the  paper, having a continual learning setup, where DEB is continuously trained on the distribution of games produced by the mechanism generator to prevent the distribution shift is one of the primary research directions to strengthen and improve those results in future work.
>
> **Baselines** We acknowledge your concern. Because of the novelty of our framework in the current form, there are no real applicable algorithms that could serve as a good baseline across the diverse tasks. We therefore actively looked for setting-specific baselines. However, most of the works mentioned for instance in Section 5.3 on multi-agent contract design make very restrictive assumptions (linear contracts, binary action spaces) and focus on approximation and complexity results but do not present implementable algorithms.
> To ground our results, we thus turned to standard optimization techniques. First, we tried grid search in combination with a cvxpy equilibrium solver for the lower level. However, this simply does not scale to the high dimensionality of the games we consider (up to 16x16 action spaces). It is thus not an informative baseline and not included in the paper. We instead opted for the “local polishing” approach that reliably acts as a lower bound on the gap between our solution and the (intractable to compute) global optimum.
>
> **Bounding performance in terms of DEB error** For our experiments, where both the generator and the equilibrium solver are neural networks, deriving theoretical bounds on the errors will be infeasible. Therefore, in our experimental section, we evaluated the generated games both at the $\varepsilon$-ME-Eql. predicted by the DEB and the exact equilibrium computed by ECOS to approximate the error empirically.
> Leaving function approximation aside, bounding the error of the loss in incentive design problems in terms of the lower-level equilibrium gap is an interesting research direction. Such bounds have been discussed for standard bilevel optimization (see e.g. Hu et al. Contextual Stochastic Bilevel Optimization) and bilevel RL (see e.g. Chen et al. Adaptive Model Design for Markov Decision Processes). For a single context, an iterative equilibrium solver with convergence guarantees, running gradient descent on the upper level loss and assuming the loss is sufficiently well-behaved, e.g. has Lipschitz gradients etc., it should be possible to derive similar results. While this was out of scope for this initial work, we would definitely welcome such theoretical work building upon our framework.
>
> We hope this addresses your questions and helps you in your final assessment of our paper. We are happy to discuss further during the rebuttal period.

---

> > ### Author Rebuttal · Reviewer_vM2u · 2026-04-02
> >
> > I appreciate the author's responses. I don't have more questions.

---

### Decision · Program_Chairs · 2026-04-30

**Decision:**

Accept (regular)

**Comment:**

This paper takes a novel approach to automated mechanism design / differentiable economics with a focus on designing strategic interactions in normal-form settings.  The reviewers agree that the approach is novel and interesting.  There are remaining concerns about scalability and the strength of baselines in the experiments.  But given this appears to be the first attempt at this type of approach I am satisfied to leave exploring these issues to subsequent work which will have this as a baseline to compare against.